# Better autoregressive regression with LLMs via regression-aware fine-tuning

**Michal Lukasik, Zhao Meng, Harikrishna Narasimhan, Yin-Wen Chang,**
**Aditya Krishna Menon, Felix X. Yu, Sanjiv Kumar**
{mlukasik,mengzhao,hnarasimhan,yinwen,adityakmenon,felixyu,sanjivk}@google.com
Google Research

## Abstract

Decoder-based large language models (LLMs) have proven highly versatile, with remarkable successes even on problems ostensibly removed from traditional language generation. One such example is solving *regression* problems, where the targets are real numbers rather than textual tokens. A common approach to use LLMs on such problems is to perform fine-tuning based on the cross-entropy loss, and use autoregressive sampling at inference time. Another approach relies on fine-tuning a separate predictive head with a suitable loss such as squared error. While each approach has had success, there has been limited study on principled ways of using decoder LLMs for regression. In this work, we compare different prior works under a unified view, and introduce *regression-aware fine-tuning* (*RAFT*), a novel approach based on the Bayes-optimal decision rule. We demonstrate how RAFT improves over established baselines on several benchmarks and model families.

## 1 Introduction

Decoder-based large language models (LLMs) (Brown et al., 2020; OpenAI et al., 2023; Anil et al., 2023; Touvron et al., 2023; Gemini Team et al., 2024; Grattafiori et al., 2024; DeepSeek-AI et al., 2024) have set new benchmarks in challenging *generative* tasks (e.g., summarization, translation, open-ended dialogue). Such models' versatility has further prompted their exploration for classic *predictive* tasks (e.g., classification, regression, ranking) (Liu & Low, 2023; Fernandes et al., 2023; Qin et al., 2023; Vacareanu et al., 2024b; Yang et al., 2023; Dukić & Snajder, 2024; Lukasik et al., 2024; Vacareanu et al., 2024a), once the purview of *encoder-only* models such as BERT (Devlin et al., 2019). Such exploration is expected to increase given the sustained efforts towards building ever-larger decoder-based LLMs, with limited parallels in scaling encoder-based models.

Our interest is in the predictive task of *natural language regression*, where the goal is to predict a real-valued target given a textual input. This covers important practical applications such as semantic similarity prediction (Cer et al., 2017), automatic quality assessment of translation (Kocmi & Federmann, 2023) or written text (Chiang & Lee, 2023), and sentiment analysis (Zhang et al., 2024). At first glance, it is not apparent how to perform numeric prediction via a model operating on textual tokens. Existing works have successfully followed two broad approaches. *Autoregressive regression* approaches directly predict as *text* the numeric targets (e.g., predict $12.34$ by iteratively predicting tokens: '1', '2', '.', '3', '4') or corresponding discretized categories (e.g., predict one of { "bad", "ok", "good" }) (Fernandes et al., 2023), via either standard autoregressive decoding (Liu & Low, 2023; Yang et al., 2023) or suitable modifications (Gruver et al., 2023; Lukasik et al., 2024; Requeima et al., 2024). Such approaches have been explored for both in-context learning, as well as standard cross-entropy loss fine-tuning (c.f. Figure 1(a)). *Predictive head* approaches side-step autoregressive decoding entirely, and instead learn a separate head on representations derived from the inputs. Common representations include mean-pooled output embeddings (Zhuang et al., 2023), and the final-position logit for a special token (e.g., <extra_id_0> in T5) (Fernandes et al., 2023).

Both autoregressive and predictive head approaches have proven successful for natural language regression tasks. However, there has been (to our knowledge) no systematic comparison between these methods; further, each of them has a conceptual shortcoming. The autoregressive regression approach does not exploit the numerical nature of the regression targets, and thus does not consider

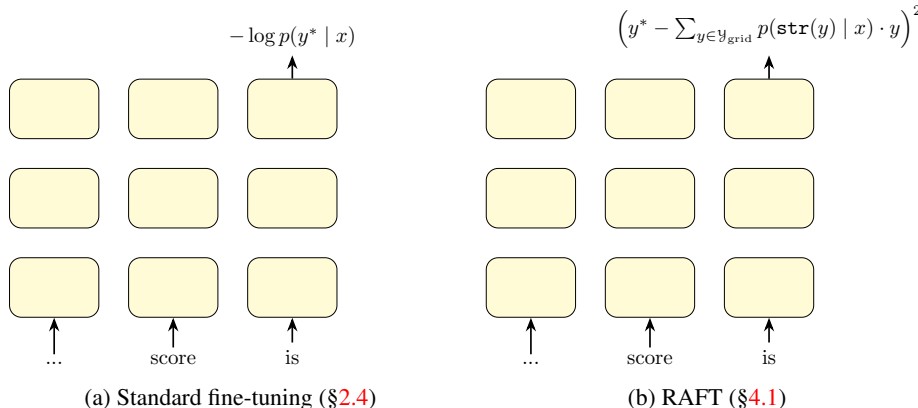

Figure 1: Illustrative example contrasting approaches to autoregressive regression fine-tuning for a decoder-only Transformer. We consider an input string $x$ ending with tokens { 'score', 'is' }, and a numerical target $y^*$, which for simplicity we assume to comprise a single string token. With abuse of notation, we use $y^*$ as both its numerical value and the corresponding string token. (a) Standard (CE) fine-tuning optimizes the likelihood $-\log p(y^* \mid x)$ of individual tokens in the numerical string representation, essentially treating numeric generation as a classification task. In general, $y^*$ can have a multi-digit representation (i.e., $y_1^* y_2^* \cdots y_k^*$), in which case the CE loss is applied over each token. (b) RAFT leverages the LLM's predictive distribution over the next numerical token (from some grid $\mathcal{Y}_{\text{grid}}$) to compute an expected numerical value, and minimizes the squared difference to the target.

the fact that for a target of 1, predicting 11 is worse than predicting 1.1. On the other hand, the predictive head approach deviates from the pre-training objective typically used in decoder-based LLMs, *viz.* next-token prediction (Radford et al., 2018), and thus may not use the model in an optimal manner. This prompts us to ask: *how can we respect both the LLM pre-training objective and the numerical nature of targets for natural language regression tasks?*

In this work, we introduce *regression-aware fine-tuning* (*RAFT*), a novel approach to autoregressive regression which makes use of the numerical nature of the targets. We prove theoretical limitations of established alternative approaches to autoregressive regression, and prove that RAFT mitigates them. We systematically compare RAFT against autoregressive and predictive head baselines, and consider several ablations for understanding the crucial design decisions for making a decoder-based LLM work under different settings. See Table 1 for an overview of both the previous works and the approach introduced in this work. Overall, our contributions are as follows:

 (i)  We identify theoretical limitations of standard approaches to LLM-based regression (§3).
 (ii) We propose regression-aware fine-tuning (RAFT), a novel approach to autoregressive regression, and prove that it mitigates the theoretical limitations of prior works (§4.1). We further present a unified view of decoder-based LLM regression approaches, capturing both the autoregressive and the prediction head approaches (§4.2).
(iii) We systematically compare autoregressive regression, predictive head and RAFT approaches across multiple datasets and LLMs, and consistently find RAFT to be the most performant. We further attempt to pinpoint the sources of differences in the performance between different approaches, explicating the design choices behind the effectiveness of RAFT (§5).

## 2 BACKGROUND

We first introduce notation and review previous works on applying decoder-based LLMs to regression.

### 2.1 NOTATION AND PROBLEM SETTING

For a finite vocabulary $\mathcal{V}$ of *tokens* (e.g., words in English), let $\mathcal{X} \subset \mathcal{V}^*$ be a set of *inputs* comprising strings of tokens, and $\mathcal{Y} \subset \mathbb{R}$ be a set of real-valued *targets*. We restrict our consideration to targets with finite (base-10) bit representations, thus excluding irrational numbers. We further assume that each $y \in \mathcal{Y}$ has a unique string representation $\mathtt{str}(y) \in \mathcal{V}^*$; for example, the integer 1 has the string encoding "1". Let $\mathbb{P}$ denote a ground-truth distribution over $\mathcal{X} \times \mathcal{Y}$, with the decomposition

| Approach | Autoregressive | Fine-tuning | Inference | References |
|---|---|---|---|---|
| Zero-shot decoding | ✓ | None | Standard decoding | Kocmi & Federmann (2023) |
| Fine-tuning and decoding | ✓ | Cross-entropy | Standard decoding | Fernandes et al. (2023) |
| Zero-shot RAIL | ✓ | None | Regression-aware decoding | Lukasik et al. (2024) |
| Fine-tuning and RAIL | ✓ | Cross-entropy | Regression-aware decoding | *this work* |
| Predictive head | ✗ | Target metric | Point estimate | Fernandes et al. (2023) |
| RAFT | ✓ | Target metric | Regression-aware decoding | *this work* |

Table 1: Summary of the approaches to applying decoder-based LLMs to natural language regression tasks. Previous works rely either on using the model autoregressively (i.e., analogously to how it was pre-trained) or as an encoder (i.e., an output is constructed based on embeddings or logits obtained for the inputs). Different training and inference approaches have been considered for both approaches.

$\mathbb{P}(x, y) = \mathbb{P}(x) \cdot \mathbb{P}(y \mid x)$. The *natural language regression* problem involves learning a predictor $\hat{y}\colon \mathcal{X} \to \mathbb{R}$ that minimises the *mean squared error* over (input, target) pairs drawn from $\mathbb{P}$:

$$L(\hat{y}) = \mathbb{E}_{(x,y^*)\sim\mathbb{P}}\left[(y^* - \hat{y}(x))^2\right].$$

The mean squared error is a canonical choice in regression problems (Fernandes et al., 2023). The *Bayes-optimal predictor* minimizing the above is $\hat{y}(x) = \mathbb{E}_{y^*\sim\mathbb{P}(\cdot|x)}[y^*]$. Classical natural language regression problems include sentiment analysis and semantic similarity prediction. More broadly, many problems can be framed as natural language regression tasks, e.g., by crafting a suitable prompt that summarizes a raw feature vector into a natural language description (Requeima et al., 2025).

We seek to employ *large language models* (LLMs) for such regression tasks. An LLM specifies a distribution $p$ over strings in $\mathcal{V}^*$. Given an input $x \in \mathcal{X}$, let $p(\cdot \mid x)$ denote the corresponding conditional distribution over possible continuations. Note that it may be possible for $p(z \mid x) > 0$ where $z \in \mathcal{V}^*$ does not have a numerical representation; we discuss this issue more in Section 2.2.

LLMs are typically *pre-trained* on large corpora via self-supervised objectives (Radford et al., 2018), and can perform *few-shot* or *in-context learning* given suitably crafted prompts (Brown et al., 2020). For example, if the goal is to predict the probability that a user will enjoy a movie titled "Cure", we may construct an input $x$ = "Hereditary: 0.7 | Ringu: 0.9 | Cure: ", and probe the LLM's estimate of plausible continuations via $p(\cdot \mid x)$. We next discuss *inference* (or *decoding*) procedures for deriving a predictor $\hat{y}$ given a pre-trained LLM.

## 2.2 STANDARD LLM INFERENCE FOR REGRESSION

Standard LLM inference involves predicting numerical targets in a generative manner, by performing autoregressive decoding to draw a sample from the distribution $p(\cdot \mid x)$:

$$\begin{aligned} \hat{y}_{\mathrm{AR}}(x) &\doteq \texttt{float}(z) \\ z &\sim p(\cdot \mid x). \end{aligned} \tag{1}$$

Here, $z \in \mathcal{V}^*$ is generated autoregressively on a token-by-token basis. Further, $\texttt{float}(z)$ denotes an operator that converts a given string $z$ (e.g., "12.34") to a corresponding numeric value (e.g., 12.34); if $z$ does not have a numeric representation (e.g., "banana"), then we assume that a suitable default value is returned. Unless otherwise stated, we assume $\texttt{float}(z) = 0.0$ for $z \notin \mathcal{Y}$.

Different algorithms may be used for the autoregressive generation of $z$, e.g., greedy decoding and temperature sampling (Naseh et al., 2023). Many such algorithms seek to approximate the mode:

$$\hat{y}_{\mathrm{mode}}(x) := \arg\max_{y\in\mathcal{Y}} p(y \mid x). \tag{2}$$

Note that one may also forcibly restrict the decoding output to comprise numerical targets, e.g., by employing a form of constrained decoding (Geng et al., 2023). However, in practice, the targets from high-quality LLMs tend to be numerical even under zero-shot settings (Lukasik et al., 2024).

## 2.3 RAIL: REGRESSION-AWARE LLM INFERENCE

Recently, Lukasik et al. (2024) pointed out a limitation of decoding the most likely target when employing autoregressive models for regression. Decoding of the most likely targets can be shown

to minimize the 0-1 loss $\ell(y, \hat{y}) = \mathbb{1}(y \neq \hat{y})$, which may not be well aligned with the square loss of interest in regression. As a remedy, instead of autoregressive decoding per Equation 1, Lukasik et al. (2024) proposed the *regression-aware inference* (*RAIL*) method, which given a loss $\ell$ and model prediction $p(\cdot \mid x)$ estimates the Bayes-optimal minimizer of the expected loss: i.e.,

$$\hat{y}_{\text{RAIL}}(x) = \arg\min_{v \in \mathbb{R}} \mathbb{E}_{y \sim p(\cdot \mid x)} \left[\ell(\texttt{float}(y), v)\right], \tag{3}$$

where $\texttt{float}(\cdot)$ is as per the previous section. For the squared loss $\ell(y, \hat{y}) = (y - \hat{y})^2$, the optimal decision rule can be shown to take the following closed-form solution:

$$\hat{y}_{\text{RAIL}}(x) = \mathbb{E}_{y \sim p(\cdot \mid x)} \left[\texttt{float}(y)\right]. \tag{4}$$

Since $p(\cdot \mid x)$ is a distribution over *all possible strings*, it is typically intractable to compute the above expectation exactly; this remains true even if we restrict attention to those strings corresponding to a valid numerical value (of which there are infinitely many). In practice, Equation 4 can be estimated either via sampling a finite number of $y$ values, or via scoring of targets. In the latter, suppose we have some restricted target grid $\mathcal{Y}_{\text{grid}} \subset \mathcal{Y}$. Then, the RAIL predictor is averaged over $\mathcal{Y}_{\text{grid}}$, yielding:

$$\hat{y}_{\text{RAIL}}(x; \mathcal{Y}_{\text{grid}}) = \sum_{y \in \mathcal{Y}_{\text{grid}}} p(\texttt{str}(y) \mid x) \cdot y. \tag{5}$$

Note that $\sum_{y \in \mathcal{Y}_{\text{grid}}} p(\texttt{str}(y) \mid x) \neq 1$ is possible, so the above is technically not an expectation; however, in practice, high-quality LLMs tend to concentrate most mass on numerical targets.

There are several choices of $\mathcal{Y}_{\text{grid}}$ available to the practitioner. For discrete targets $\mathcal{Y}$ of moderate size, one may simply choose $\mathcal{Y}_{\text{grid}} = \mathcal{Y}$. For bounded $\mathcal{Y}$, one choice is equally spaced targets covering the range of $\mathcal{Y}$, e.g. integers or fixed-precision numbers (e.g. 2 decimal points) (Lukasik et al., 2024).

## 2.4 STANDARD FINE-TUNING FOR REGRESSION

The above approaches operate on a pre-trained LLM via few-shot prompting. However, it has been consistently observed that direct *fine-tuning* of LLMs on the task of interest can be beneficial (Liu et al., 2022). Fine-tuning seeks to adapt an LLM to the target distribution $\mathbb{P}$ by minimizing

$$L(p) = \mathbb{E}_{(x, y^*) \sim \mathbb{P}} \left[\ell(y^*, p(\cdot \mid x))\right] \tag{6}$$

for a suitable *loss function* $\ell \colon \mathcal{Y} \times \Delta_{\mathcal{V}^*} \to \mathbb{R}$, where $\Delta_S$ denotes the set of distributions over a set $S$. Given a sample $S \in (\mathcal{X} \times \mathcal{Y})^N$ of $N$ (input, target) pairs drawn from $\mathbb{P}$, the empirical loss is

$$\hat{L}(p) = \frac{1}{N} \sum_{(x, y^*) \in S} \ell(y^*, p(\cdot \mid x)). \tag{7}$$

A standard choice of $\ell$ is the log-loss (also referred to as cross-entropy):

$$\ell(y^*, p(\cdot \mid x)) = -\log p(\texttt{str}(y^*) \mid x), \tag{8}$$

recalling that $\texttt{str}(y^*)$ denotes the string representation of a numeric target $y^* \in \mathbb{R}$. More generally, one may use categorical descriptions of the target after discretising to some finite grid $\mathcal{Y}_{\text{grid}} \subset \mathcal{Y}$; e.g., { "very bad", "bad", "ok", "good", "very good" } (Fernandes et al., 2023).

A model obtained from standard fine-tuning can rely on either of the above decoding procedures (standard decoding §2.2 or RAIL §2.3) at inference time.

## 2.5 PREDICTIVE HEAD APPROACHES TO REGRESSION

Predictive head approaches are an alternative to standard fine-tuning. Here, one constructs a predictor $\hat{y}(x)$ by utilizing activations or embeddings from a forward pass of the LLM. Abstractly, we first extract an *input representation* $\Phi(x) \in \mathbb{R}^q$, which is then fed into a *regressor* $s \colon \mathbb{R}^q \to \mathbb{R}$. Canonically, the regressor is a linear model, yielding $\hat{y}(x) = b + w^\top \Phi(x)$ for learnable $w \in \mathbb{R}^q, b \in \mathbb{R}$.

Various choices for $\Phi(x)$ have been considered in previous works. To describe these, we need some additional notation. Let $V \doteq |\mathcal{V}|$. Given a string $x \in \mathcal{V}^*$ of length $L$, a *Transformer-based* language

model (Vaswani et al., 2017) first constructs an input embedding $\epsilon_{\text{in}}(x) = E_{\text{in}}\epsilon_{\text{oh}}(x) \in \mathbb{R}^{D \times L}$, where $E_{\text{in}} \in \mathbb{R}^{D \times V}$ is a matrix of $D$-dimensional token embeddings, and $\epsilon_{\text{oh}}(x) \in \mathbb{R}^{V \times L}$ is the one-hot embedding of each token in $x$. Next, this input embedding is passed through a stack of attention and MLP layers, to produce the output embedding $\epsilon_{\text{out}}(x) \in \mathbb{R}^{D \times L}$. One further projects this to the vocabulary space to produce output logits $f_{\text{out}}(x) = E_{\text{out}}^{\top}\epsilon_{\text{out}}(x) \in \mathbb{R}^{V \times L}$, where $E_{\text{out}} \in \mathbb{R}^{D \times V}$. Finally, one transforms these to a distribution $p_{\text{out}}(\cdot \mid x) = \text{softmax}(f_{\text{out}}(x)) \in [0,1]^{V \times L}$ over possible tokens via the softmax operator. For certain models (e.g., Gemma), one ties $E_{\text{in}} = E_{\text{out}}$.

Given the above, one may extract an input representation through multiple means, most commonly *pooling* or selection of the *output token embeddings*, *output logits*, or *output probabilities*. For example, we may consider the final-token logit activation for a special token $v_* \in \mathcal{V}$ (Fernandes et al., 2023; Zhuang et al., 2023), or mean-pooling the output token embeddings (Zhuang et al., 2023).

Given a suitable predictor, one may directly optimize the mean squared error during fine-tuning. Note here that no autoregressive decoding is conducted at inference.

# 3 LIMITATIONS OF STANDARD FINE-TUNING FOR REGRESSION

We analyze limitations of standard fine-tuning for autoregressive regression (proofs in Appendix A).

## 3.1 LIMITATIONS OF STANDARD FINE-TUNING AND STANDARD DECODING

A natural baseline is to employ cross-entropy based fine-tuning by minimizing Equation 7, and to then apply standard decoding (see Equation 2). Since the log-loss is strictly proper, minimizing Equation 7 recovers the Bayes distribution $\mathbb{P}(\cdot \mid x)$ in the population limit (Gneiting & Raftery, 2007). In practice, however, the fine-tuned model distribution $p(\cdot \mid x)$ may deviate from $\mathbb{P}(\cdot \mid x)$. The following Lemma shows that even when the fine-tuned model distribution $p(\cdot \mid x)$ perfectly fits $\mathbb{P}$, the standard decoding predictor can incur a high squared error compared to the Bayes-optimal predictor.

**Lemma 1.** Assume $|\mathcal{Y}| \geq 2$ and $0 \in \mathcal{Y}$, with $N \doteq \max(\mathcal{Y})$. For any $\epsilon \in [0, 0.5]$, there exist $\mathbb{P}, p$ such that: $\mathbb{E}_x\left[\mathbb{D}_{\text{KL}}\big(\mathbb{P}(\cdot|x), p(\cdot|x)\big)\right] \leq \epsilon$, and $\mathbb{E}_x\left[\big(\mathbb{E}_{y^* \sim \mathbb{P}(\cdot|x)}[y^*] - \hat{y}_{\text{mode}}(x)\big)^2\right] \geq \left(\frac{N}{2}\right)^2$.

Thus, using cross-entropy fine-tuning with standard decoding is not well-aligned with the eventual goal of approximating $\mathbb{E}_{y^* \sim \mathbb{P}(\cdot|x)}[y^*]$. Simply stated, although the distance between a predicted probability distribution $p$, and the true distribution $\mathbb{P}$ may be small (or even 0), the squared error between the mean of the distribution $\mathbb{P}$ and the mode of the distribution $p$ can be disproportionately large. This is because the mode of a distribution can be far from its mean, leading to a high error.

## 3.2 LIMITATIONS OF STANDARD FINE-TUNING AND RAIL DECODING

Given that the standard decoding can lead to arbitrary large errors due to the predictor being ill suited for regression tasks, one may expect a better approach would be to employ cross-entropy based fine-tuning by minimizing Equation 7, and to then apply the RAIL decoding (see Equation 5). This approach indeed mitigates the issue of standard decoding being misaligned with the Bayes-optimal predictor. However, we can show that when the fine-tuned model distribution $p(\cdot \mid x)$ deviates from $\mathbb{P}$ by even a small error, the predictor can lead to a significant squared error.

**Lemma 2.** Assume $|\mathcal{Y}| \geq 2$ and $0 \in \mathcal{Y}$, with $N \doteq \max(\mathcal{Y})$. For any $\epsilon \in [0, 0.5]$, there exists $\mathbb{P}, p$ such that: $\mathbb{E}_x\left[\mathbb{D}_{\text{KL}}\big(\mathbb{P}(\cdot|x), p(\cdot|x)\big)\right] \leq \epsilon$, and $\mathbb{E}_x\left[\big(\mathbb{E}_{y^* \sim \mathbb{P}(\cdot|x)}[y^*] - \hat{y}_{\text{RAIL}}(x)\big)^2\right] \geq \left(\frac{N}{4}\right)^2 \epsilon$.

Thus again, using cross-entropy fine-tuning with RAIL might not align with the goal of approximating $\mathbb{E}_{y^* \sim \mathbb{P}(\cdot|x)}[y^*]$. Intuitively, cross-entropy fine-tuning treats all "wrong" predictions the same, as it is unaware of the difference in the magnitude of the numerical values represented by the tokens; e.g., if 100 and 1,000 are two incorrect predictions each represented by single tokens, then placing a large mass on the token representing 100 is penalized similarly to placing a large mass on the token 1,000.

One solution to the above issue is to *directly employ the RAIL predictor in the fine-tuning process*. This requires going beyond the log-loss in Equation 8, as we detail next.

## 4 REGRESSION-AWARE AUTOREGRESSIVE LLM TRAINING

To overcome the drawbacks of using RAIL with traditional fine-tuning, we propose a novel regression-aware objective that directly minimizes the squared loss on the RAIL predictor.

### 4.1 REGRESSION-AWARE LLM FINE-TUNING

As an alternative to the standard log-loss, we propose the following.

**Definition 1.** Define the *regression-aware fine-tuning* (RAFT) loss as follows:

$$\ell_{\mathrm{RAFT}}(y^*, p(\cdot \mid x)) = \left(y^* - \mathbb{E}_{y \sim p(\cdot \mid x)}[\mathtt{float}(y)]\right)^2. \tag{9}$$

Equally, this uses the RAIL predictor $\hat{y}_{\mathrm{RAIL}}(x)$ to construct a numeric value from the LLM, and measures the square loss against the target $y^*$. Given a finite grid $\mathcal{Y}_{\mathrm{grid}} \subset \mathcal{Y}$ and fine-tuning set $S$, the empirical loss corresponding to Equation 9 is:

$$\hat{L}_{\mathrm{RAFT}}(p; \mathcal{Y}_{\mathrm{grid}}) = \frac{1}{N} \sum_{(x,y^*) \in S} \left( y^* - \sum_{y \in \mathcal{Y}_{\mathrm{grid}}} p(\mathtt{str}(y) \mid x) \cdot y \right)^2. \tag{10}$$

Note that computing this loss only requires scoring each $y \in \mathcal{Y}_{\mathrm{grid}}$ under the model; we do not need to perform any explicit sampling or decoding during training.

Compared to standard fine-tuning with RAIL decoding, we attempt to avoid the issue in Lemma 2 by seeking to *directly* minimize $\mathbb{E}_x\left[\left(\mathbb{E}_{y^* \sim \mathbb{P}(\cdot \mid x)}[y^*] - \hat{y}_{\mathrm{RAIL}}(x)\right)^2\right]$. Surprisingly, despite computing $\hat{y}_{\mathrm{RAIL}}(x)$ over the restricted target space $\mathcal{Y}_{\mathrm{grid}}$, under mild conditions the minimizer of Equation 9 exactly mimics the Bayes-optimal predictor over the full space $\mathcal{Y}$.

**Lemma 3.** Suppose $\mathcal{Y} \subset \mathbb{R}$, and $\arg \min_{y \in \mathcal{Y}} \in \mathcal{Y}_{\mathrm{grid}}$ and $\arg \max_{y \in \mathcal{Y}} \in \mathcal{Y}_{\mathrm{grid}}$. Let $p^*(\cdot \mid x)$ be the minimizer of the RAFT loss from Definition 1 over all distributions $p(\cdot \mid x)$. Then the RAIL predictor $\hat{y}_{\mathrm{RAIL}}(x; \mathcal{Y}_{\mathrm{grid}}) = \sum_{y \in \mathcal{Y}_{\mathrm{grid}}} p^*(\mathtt{str}(y) \mid x) \cdot y$ constructed from $p^*(\cdot \mid x)$ satisfies:

$$\hat{y}_{\mathrm{RAIL}}(x; \mathcal{Y}_{\mathrm{grid}}) = \mathbb{E}_{y^* \sim \mathbb{P}(\cdot \mid x)}[y^*].$$

The intuition behind this result is that any numerical target in $\mathcal{Y}$ can be expressed by a convex combination of the smallest and largest numbers in $\mathcal{Y}$, and can thus be realized by the RAIL predictor.

In Appendix C we contrast RAFT loss against a variant optimizing regression metrics via sampled model predictions, following the Minimum Bayes Risk prediction literature (Kaiser et al., 2000; Shannon, 2017; Prabhavalkar et al., 2018).

### 4.2 CONTRASTING AND UNIFYING RAFT AND PREDICTIVE HEAD APPROACHES

Our discussion of RAFT highlighted its close relation to autoregressive RAIL decoding, which appears rather different to predictive head approaches. However, in the case of a *single-digit* grid $\mathcal{Y}_{\mathrm{grid}}$ (wherein each element corresponds to a single token in $\mathcal{V}$), the predictor function for RAFT bears similarities to the predictive head approaches. Note that if $y \in \mathcal{Y}_{\mathrm{grid}}$ corresponds to a single token, by definition $p(\mathtt{str}(y) \mid x) = p_{\mathrm{out}}(\cdot \mid x)_{\mathtt{str}(y), L}$. Then, the RAIL predictor becomes

$$\hat{y}_{\mathrm{RAIL}}(x; \mathcal{Y}_{\mathrm{grid}}) = \sum_{y \in \mathcal{Y}_{\mathrm{grid}}} y \cdot p(\mathtt{str}(y) \mid x) = \sum_{y \in \mathcal{Y}_{\mathrm{grid}}} y \cdot p_{\mathrm{out}}(\cdot \mid x)_{\mathtt{str}(y), L}.$$

We may now contrast RAFT against various predictors considered in prior work (see Table 2 for a summary). In particular, it is instructive to compare this with the final-token logit activation method from Table 2. Both take the following form for an activation $\Psi$ and weight vector $w \in \mathbb{R}^{\mathcal{V}}$:

$$\hat{y}(x) = b + w^\top \Psi\left(f_{\mathrm{out}}(x)_{:,L}\right).$$

Contrasting the RAFT and the final-token logit method, we observe the following differences:

| Category | Approach | Predictor function $\hat{y}(x)$ | Fine-tuning loss |
|---|---|---|---|
| Autoregressive (prior works) | Standard decoding zero-shot (Kocmi & Federmann, 2023) | $\arg\max_{y \in \mathcal{Y}} p(y \mid x)$ | N/A |
| | RAIL zero-shot (Lukasik et al., 2024) | $\sum_{y' \in \mathcal{Y}} y' \cdot p(y'|x)$ | N/A |
| | Standard fine-tuning and standard decoding (Fernandes et al., 2023) | $\arg\max_{y \in \mathcal{Y}} p(y \mid x)$ | $-\log p(y^* \mid x)$ |
| Autoregressive (this work) | RAIL standard fine-tuning | $\sum_{y' \in \mathcal{Y}} y' \cdot p(y'|x)$ | $-\log p(y^* \mid x)$ |
| RAFT (this work) | General RAFT | $\sum_{y' \in \mathcal{Y}} y' \cdot p(y'|x)$ | $(\hat{y}(x) - y^*)^2$ |
| | Single-digit RAFT | $\sum_{y' \in \mathcal{Y}_{digit}} y' \cdot p_{\text{out}}(x)_{y',L}$ | $(\hat{y}(x) - y^*)^2$ |
| Predictive head (prior works) | Final-token logit (Fernandes et al., 2023) | $b + f_{\text{out}}(x)_{v^*,L}$ | $(\hat{y}(x) - y^*)^2$ |
| | Pooled output embeddings (Zhuang et al., 2023) | $b + w^\top \text{pool}(\epsilon_{\text{out}}(x))$ | $(\hat{y}(x) - y^*)^2$ |
| Predictive head (this work) | MLP on the final-token logits | $b + \text{MLP}(E_{\text{out}}^\top \epsilon_{\text{out}}(x)_{:,L})$ | $(\hat{y}(x) - y^*)^2$ |
| | Probability-vector projection | $b + w^\top p_{\text{out}}(x)_{:,L}$ | $(\hat{y}(x) - y^*)^2$ |
| | Learnable regression-aware training | $\sum_{y' \in \mathcal{Y}} w_{y'} \cdot p_{\text{out}}(x)_{y',L}$ | $(\hat{y}(x) - y^*)^2$ |

Table 2: Approaches for applying decoder-based LLMs to regression. Here, $p(\cdot \mid x)$ denotes a distribution over possible outputs given an input string $x$, and $\hat{y}(x) \in \mathbb{R}$ a predictor given by a predictive head approach. $b$ and $w$ are learnable parameters, $v^* \in \mathcal{V}$ is a fixed token, pool is a pooling operator, $L$ is the length of input $x$, and $\mathcal{Y}_{\text{digits}}$ denotes all digits covering the range of targets (unless otherwise stated, '1'–'5'). The first 4 rows show the autoregressive baselines: standard decoding (Section 2.2), RAIL zero-shot (Section 2.3), standard fine-tuning and decoding (Section 2.4), RAIL with standard fine-tuning (Section 3.2). The next 2 rows show RAFT: the general autoregressive form ($\mathcal{Y} = \mathcal{Y}_{\text{grid}}$ for general output spaces), and the single digit version (e.g. $\mathcal{Y} = \{1, 2, 3, 4, 5\}$). The following 2 rows present the prior works from Fernandes et al. (2023); Zhuang et al. (2023). The last 3 rows present new predictive head approaches that attempt to mimic the behavior of RAFT.

- *Activation*: for single-digit RAFT, $\Psi$ is the softmax activation that converts $f_{\text{out}}(x)$ to the probability vector $p_{\text{out}}(\cdot \mid x)$. For the final-token logit, $\Psi$ is the identity activation.
- *Weight vector*: for the final-token logit, $w$ is a one-hot vector with 1 corresponding to the special token position. For single-digit RAFT, $w_v = \text{float}(v)$ for each $v \in \mathcal{V}$; note that, as a result, positions corresponding to non-digits have weight 0.
- *Initialization*: another important factor is that the RAFT predictor at initialization exactly coincides with RAIL decoding, and thus, forms a strong predictor for zero-shot inference with LLMs (Lukasik et al., 2024). By contrast, most predictive head approaches will incur a high error at initialization due to deviating from the next token prediction task. Therefore, RAFT can be seen as a predictive head approach with strong performance at initialization, potentially making optimization easier; indeed, we empirically observe RAFT to converge faster compared to the baselines (see Figure 4 and Figure 5 in Appendix E.7).

In light of the close similarities between RAFT and the final-token logit approach, it is prudent to carefully analyze these differences and identify whether any of these choices play an important role in RAFT's performance. Therefore, we introduce the following new predictive head variants:

- *MLP on final-token logits*: this is a variant of the final-token logit method, wherein a 2-layer MLP with a non-linear activation (sigmoid) is employed on the entire final-token logit vector, rather than selecting the logit for a single special token:

$$\hat{y}(x) = b + \text{MLP}(f_{\text{out}}(x)_{:,L}). \tag{11}$$

- *Learnable-RAFT*: this is a variant of RAFT, wherein the weights over the output model probabilities are learned, rather than being fixed to the vector $w_v = \text{float}(v)$:

$$\hat{y}(x) = \sum_{y' \in \mathcal{Y}} w_{y'} \cdot p_{\text{out}}(x)_{\text{str}(y'),L} \tag{12}$$

The learnable-RAFT variant adds more flexibility to the predictor function $\hat{y}$ over the vanilla RAFT method. However, as with other predictive head methods, it deviates from the next-token prediction pre-training task. Which of these two factors — predictor flexibility, and alignment to the pre-training task — is the most important? To answer this question, we compare learnable-RAFT against RAFT, and also experiment with fine-tuning from a *randomly initialized* (as opposed to a pre-trained) model.

| Dataset | Model size | Zero-shot standard decoding | Zero-shot RAIL | Standard fine-tuning standard decoding | Standard fine-tuning RAIL | Predictive head | RAFT |
|---|---|---|---|---|---|---|---|
| Wireless | 2B | 0.88 | 0.88 | $0.70 \pm 0.01$ | $0.67 \pm 0.01$ | $0.51 \pm 0.01$ | $\mathbf{0.47 \pm 0.01}$ |
| (Amazon) | 9B | 0.89 | 0.88 | $0.78 \pm 0.05$ | $0.86 \pm 0.03$ | $0.46 \pm 0.00$ | $\mathbf{0.45 \pm 0.00}$ |
| Personal care | 2B | 1.00 | 0.97 | $0.77 \pm 0.01$ | $0.74 \pm 0.01$ | $0.52 \pm 0.02$ | $\mathbf{0.49 \pm 0.00}$ |
| (Amazon) | 9B | 0.97 | 0.95 | $0.73 \pm 0.14$ | $0.59 \pm 0.01$ | $0.48 \pm 0.01$ | $\mathbf{0.47 \pm 0.01}$ |
| Music | 2B | 1.30 | 1.29 | $1.16 \pm 0.11$ | $0.88 \pm 0.12$ | $0.52 \pm 0.00$ | $\mathbf{0.50 \pm 0.00}$ |
| (Amazon) | 9B | 1.29 | 1.29 | $0.83 \pm 0.35$ | $0.61 \pm 0.02$ | $0.50 \pm 0.00$ | $\mathbf{0.47 \pm 0.00}$ |
| STSB | 2B | 1.10 | 1.05 | $0.59 \pm 0.01$ | $0.57 \pm 0.00$ | $\mathbf{0.54 \pm 0.00}$ | $\mathbf{0.54 \pm 0.01}$ |
|  | 9B | 1.37 | 1.29 | $0.58 \pm 0.00$ | $0.53 \pm 0.00$ | $0.52 \pm 0.00$ | $\mathbf{0.51 \pm 0.01}$ |

Table 3: RMSE across datasets, methods, and Gemma-2 models of varying sizes. Fine-tuning methods report mean $\pm$ std dev from model retraining. See Table 10 (Appendix) for Gemma-2 27B.

## 5 EXPERIMENTAL RESULTS

We now present experiments and ablations comparing the autoregressive regression, predictive head and RAFT approaches on natural language regression datasets. We make the following main empirical findings: (i) RAFT outperforms all autoregressive regression and predictive head baselines across datasets and models; (ii) ablations indicate the importance of aligning fine-tuning to the pre-training loss; (iii) RAFT tends to work well when the grid $\mathcal{Y}_{\text{grid}}$ corresponds to digit tokens.

### 5.1 EXPERIMENT SETTINGS

**Datasets.** We use the following natural language regression datasets, with RMSE as the main metric:

(i) *US Amazon reviews*, where we aim to predict the 5-star rating for a product review (Ni et al., 2019). We consider a few categories from the Amazon reviews datasets, each forming a separate dataset: Wireless, Music, Personal products. We use 1,500 examples for the test set (after Lukasik et al. (2024)), 1,500 for validation and 10,000 examples for training.

(ii) *Semantic Textual Similarity Benchmark* (*STSB*) (Cer et al., 2017), comprising of sentence pairs human-annotated with a similarity score from 0 to 5. To measure the impact of varying the dataset size, we also consider 1,000 examples for training (*STSB 1K*; see Table 8 in Appendix).

(iii) *MovieLens-1M*, where we construct a movie rating prediction task following Luo et al. (2024).

We summarize the dataset statistics and the prompts in Table 6 and Table 7 (Appendix).

**Models.** We experiment with Gemma-2 (Team et al., 2024) and PaLM-2 (Anil et al., 2023) instruction-tuned model families of different sizes (see Appendix D for training details). Where standard deviations are reported, fine-tuning is performed 3 times.

**Methods.** We compare the following methods: (1) autoregressive baselines (Section 2.2), RAIL zero-shot (Section 2.3), RAIL with cross-entropy fine-tuning (Section 3.2); (2) predictive head approaches from Fernandes et al. (2023); Zhuang et al. (2023); (3) the new RAFT method (Section 4.1); (4) new predictive head approaches that attempt to mimic the behavior of RAFT (Section 4.2). In zero-shot standard decoding, we use greedy decoding; in zero-shot RAIL, the predictor is obtained by sampling 32 targets with temperature $T = 1$ (Lukasik et al., 2024). We also run ablations with replacing causal attention masking with bi-directional attention masking, following previous works on classification with decoder-based LLMs (Dukić & Snajder, 2024; Qorib et al., 2024).

**Implementation of the RAFT objective.** An important practical consideration is the grid $\mathcal{Y}_{\text{grid}}$. Unless otherwise stated, we choose $\mathcal{Y}_{\text{grid}} = \{ 1, 2, 3, 4, 5 \}$ as targets from all considered datasets belong to $[0, 5]$. For Amazon reviews datasets, $\mathcal{Y}_{\text{grid}} = \mathcal{Y}$, while for STSB, $\mathcal{Y}_{\text{grid}} \subset \mathcal{Y}$ (as the targets take floating point values). Recall that RAFT can represent real valued targets even under such a choice for $\mathcal{Y}_{\text{grid}}$ (see Lemma 3). Nonetheless, we analyze the impact of $\mathcal{Y}_{\text{grid}}$ on the results.

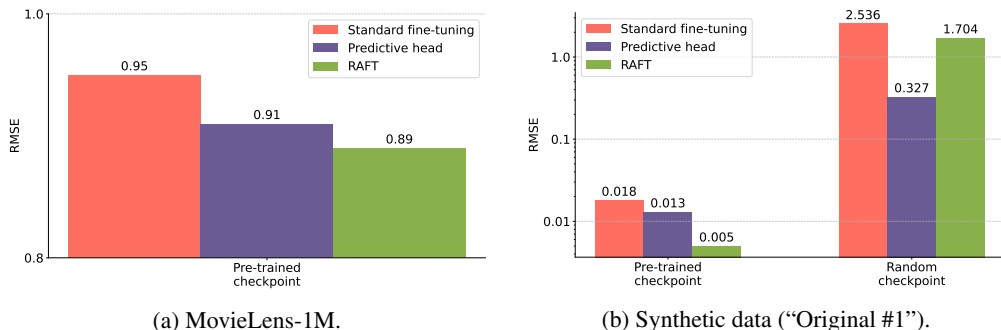

(a) MovieLens-1M.  (b) Synthetic data ("Original #1").

Figure 2: Comparing standard fine-tuning, predictive head and RAFT approaches on MovieLens (left) and the synthetic "Original #1" (right) datasets with Gemma-2 2B. RAFT outperforms the baselines on MovieLens. RAFT also improves over the baselines on the synthetic dataset "Original # 1" when using a pre-trained — but not a randomly initialized — checkpoint. (Best viewed in color.)

| Approach | Gemma-2 2B | Gemma-2 9B |
|---|---|---|
| Standard fine-tuning | 0.83 | 0.82 |
| Special-token logit (Fernandes et al., 2023) | 0.51 | 0.46 |
| Special-token logit + 2-layer MLP | 0.48 | 0.46 |
| Pooled output embeddings (mean) (Zhuang et al., 2023) | 0.50 | 0.47 |
| Pooled output embeddings (min) | 1.38 | 1.18 |
| Pooled output embeddings (max) | 1.32 | 1.13 |
| RAFT | **0.47** | **0.45** |
| Learnable-RAFT over all tokens | 0.48 | **0.45** |
| Learnable-RAFT over digits '1'–'5' | **0.47** | **0.45** |

Table 4: RMSE on Gemma-2 models on Amazon Wireless across predictive head and RAFT variants. RAFT outperforms the different predictive head variants. Additionally, we find that RAFT with learnable decision rule (*learnable-RAFT*) does not improve over RAFT with the fixed decision rule.

## 5.2 RAFT LEADS TO BETTER AUTOREGRESSIVE REGRESSION

We compare different autoregressive and prediction head approaches across across Gemma-2 models of varying sizes in Table 3. We report additional results from PaLM-2 models on STSB in Table 11 (Appendix) to verify the findings across an additional model family. We make several observations.

First, we verify the value of both (1) use of an appropriate decoding strategy (greedy versus regression-aware inference), and (2) fine-tuning over zero-shot inference. Indeed, we find that zero-shot greedy decoding, RAIL (see Section 2.3), standard fine-tuning with greedy decoding (see Section 3.2) and standard fine-tuning with RAIL inference (see Section 4.1) work increasingly better.

Second, we find that the predictive head approach outperforms the autoregressive baselines, including those that perform standard fine-tuning. This corroborates Lemma 1 and Lemma 2, which pointed at the limitations of standard fine-tuning due to it being misaligned with the squared error.

Finally, we find RAFT to outperform the predictive head and the autoregressive approaches across almost all settings. RAFT outperforming the autoregressive approaches corroborates the posited importance of aligning the fine-tuning loss in regression tasks to a regression loss. RAFT outperforming the predictive head approach corroborates the posited importance of not deviating from the autoregressive setting, which aligns with the next-token prediction pre-training task.

To further evaluate RAFT, we next contrast the key approaches (RAFT, predictive head and standard fine-tuning) on a large scale dataset for the movie recommendation problem (MovieLens-1M) with Gemma-2 2B model. We again find RAFT to improve over the baselines (see Figure 2(a)).

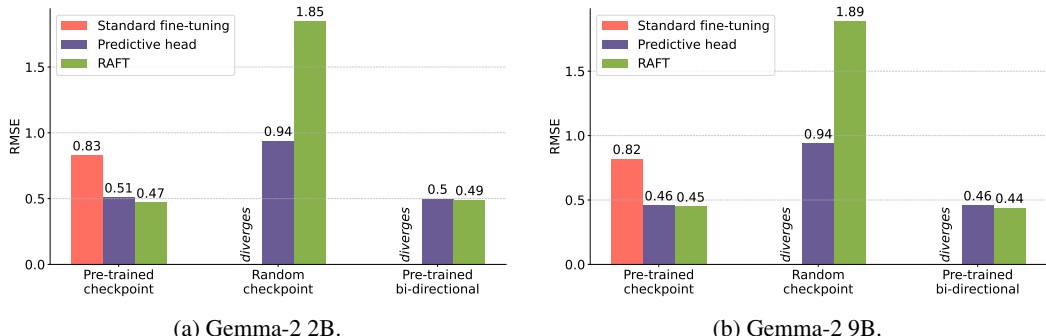

(a) Gemma-2 2B.  (b) Gemma-2 9B.

Figure 3: Comparing standard fine-tuning, predictive head and RAFT variants on Amazon Wireless. We find that changing causal to bi-directional attention masking does not significantly affect the results of the predictive head and RAFT, while expectedly making standard fine-tuning untrainable (due to the model being able to attend to the predicted target). We also analyze the effect of random initialization, and find all methods to perform worse. RAFT improves over predictive head when initializing from a pretrained checkpoint, but not when initializing from a random checkpoint, empirically corroborating the argument for better alignment of RAFT to the pre-training task.

### 5.3 Ablating design choices in RAFT and predictive head approaches

**Predictive head non-linear variant.** We experiment with two variants of learnable-RAFT (see Equation 12): one where the weight vector is learnt for all vocabulary entries, and the second, where only entries corresponding to digits '1'–'5' are learnt, while other entries are fixed to 0. For both variants we found it necessary to initialize from the solution corresponding to RAFT, as random initialization did not lead to good training dynamics. Overall, as reported in Table 4, we find both approaches to not improve over RAFT. We also consider adding a non-linear function over the special-token logit in the form of a two-layer MLP (see Equation 11). This again does not lead to improvements over RAFT. Both results demonstrate that it may be more important to align the fine-tuning to the pre-training loss, as opposed to only try make the predictor more expressive.

**Predictive head design choices.** We experiment with additional predictive head variants, including pooling over the full sequence of token embeddings from the LLM, instead of taking the final token activation. As shown in Table 4, we find it to not lead to significantly better results than the special-token logit method. Additionally, as seen on Figure 3, we find bi-directional masking of attention does not significantly affect the results of the predictive head and RAFT, while expectedly making standard fine-tuning untrainable (due to the model being able to attend to the predicted target during training). Overall, none of the predictive head variants improves over the RAFT approaches, again suggesting the importance of not deviating from the pre-training loss in fine-tuning.

**Role of pre-training.** To shed light on the importance of aligning the method with the pre-training task, we experiment with fine-tuning from a randomly initialized checkpoint (see Figure 3). We find that standard fine-tuning does not converge to a reasonable result, RAFT converges to a poor predictor, and predictive head fares the best. We next run experiments on a synthetic regression dataset from Vacareanu et al. (2024a) (the *Original #1* dataset) and report results in Figure 2(b). We corroborate the finding that RAFT improves over predictive head when initialized from a pre-trained checkpoint, and not when the model weights are initialized randomly. This supports the hypothesis that RAFT outperforms predictive head due to alignment with the pre-training task.

## 6 Discussion and future work

We introduced *regression-aware fine-tuning* (*RAFT*), a new method for fine-tuning decoder-based LLMs to predict numeric targets. We demonstrated empirically that RAFT can consistently outperform existing methods that perform standard cross-entropy fine-tuning, as well as methods that construct separate predictive heads. An interesting direction for study would be applications of such techniques to problems like time-series forecasting, as well as problems of ordinal regression.

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

# Appendix

## Table of Contents

# A PROOFS OF THEORETICAL RESULTS

## A.1 PROOF FOR LEMMA 1

**Lemma** (Restated). Assume $|\mathcal{Y}| \geq 2$ and $0 \in \mathcal{Y}$, with $N \doteq \max(\mathcal{Y})$. For any $\epsilon \in [0, 0.5]$, there exist $\mathbb{P}, p$ such that: $\mathbb{E}_x \left[ \mathbb{D}_{\mathrm{KL}} \left( \mathbb{P}(\cdot|x), p(\cdot|x) \right) \right] \leq \epsilon$, and $\mathbb{E}_x \left[ \left( \mathbb{E}_{y^* \sim \mathbb{P}(\cdot|x)} [y^*] - \hat{y}_{\mathrm{mode}}(x) \right)^2 \right] \geq \left( \frac{N}{2} \right)^2$.

*Proof.* Pick any $\alpha \doteq [0, 1]$. Recall that $N \doteq \max(\mathcal{Y})$. Consider a distribution $\mathbb{P}$ where, for any example $x \in \mathcal{X}$, $\mathbb{P}(0 \mid x) = \frac{1+0.5\alpha}{2}$, $\mathbb{P}(N \mid x) = \frac{1-0.5\alpha}{2}$, and all other targets attain probability 0.

Next, consider the model distribution: $p(0 \mid x) = \frac{1-0.5\alpha}{2}$, $p(N \mid x) = \frac{1+0.5\alpha}{2}$, and all other targets attain probability 0. Then, for any $x \in \mathcal{X}$, we have: $(\hat{y}_{\mathrm{mode}}(x) - \mathbb{E}_{y^* \sim \mathbb{P}}[y^*])^2 = \left( \frac{N}{2} \right)^2 \left( 1 + \frac{\alpha}{2} \right)^2 \geq \left( \frac{N}{2} \right)^2$ and $\|\mathbb{P}(\cdot|x) - p(\cdot|x)\|_1 = \alpha$. Using the reverse Pinsker's inequality (Sason, 2015), we further have

$$\mathbb{D}_{\mathrm{KL}} \left( \mathbb{P}(\cdot|x), p(\cdot|x) \right) \leq \log \left( 1 + \frac{\|\mathbb{P}(\cdot|x) - p(\cdot|x)\|_1^2}{\min_{y \in \mathcal{Y}} p(y|x)} \right) = \log \left( 1 + \frac{2\alpha^2}{1 - 0.5\alpha} \right) \leq \log \left( 1 + 4\alpha^2 \right),$$

where we use the fact that $\alpha \leq 1$.

Setting $\alpha = \frac{1}{2} \sqrt{e^\epsilon - 1}$, for any $\epsilon \in [0, \log(5)]$, we have:

$$\mathbb{E}_x \left[ \mathbb{D}_{\mathrm{KL}} \left( \mathbb{P}(\cdot|x), p(\cdot|x) \right) \right] \leq \epsilon; \qquad \mathbb{E}_x \left[ \left( \hat{y}_{\mathrm{mode}}(x) - \mathbb{E}_{y^* \sim \mathbb{P}}[y^*] \right)^2 \right] \geq \left( \frac{N}{2} \right)^2.$$

$\square$

## A.2 PROOF FOR LEMMA 2

**Lemma** (Restated). Assume $|\mathcal{Y}| \geq 2$ and $0 \in \mathcal{Y}$, with $N \doteq \max(\mathcal{Y})$. For any $\epsilon \in [0, 0.5]$, there exists $\mathbb{P}, p$ such that: $\mathbb{E}_x \left[ \mathbb{D}_{\mathrm{KL}} \left( \mathbb{P}(\cdot|x), p(\cdot|x) \right) \right] \leq \epsilon$, and $\mathbb{E}_x \left[ \left( \mathbb{E}_{y^* \sim \mathbb{P}(\cdot|x)} [y^*] - \hat{y}_{\mathrm{RAIL}}(x) \right)^2 \right] \geq \left( \frac{N}{4} \right)^2 \epsilon$.

*Proof.* Pick any $\alpha \in (0, 1]$. Recall that $N \doteq \max(\mathcal{Y})$. Consider a distribution $\mathbb{P}$ such that, for any example $x \in \mathcal{X}$, $\mathbb{P}(0 \mid x) = \frac{1+0.5\alpha}{2}$, $\mathbb{P}(N \mid x) = \frac{1-0.5\alpha}{2}$, and all other targets attain probability 0.

Next, consider the model distribution: $p(0 \mid x) = \frac{1-0.5\alpha}{2}$, $p(N \mid x) = \frac{1+0.5\alpha}{2}$, and all other targets attain probability 0. Then, for any $x \in \mathcal{X}$: $\left( \mathbb{E}_{y^* \sim \mathbb{P}(\cdot|x)} [y^*] - \hat{y}_{\mathrm{RAIL}}(x) \right)^2 = \left( \frac{\alpha N}{2} \right)^2$ and $\|\mathbb{P}(\cdot|x) - p(\cdot|x)\|_1 = \alpha$. Using the reverse Pinsker's inequality (Sason, 2015), we further have

$$\mathbb{D}_{\mathrm{KL}} \left( \mathbb{P}(\cdot|x), p(\cdot|x) \right) \leq \log \left( 1 + \frac{\|\mathbb{P}(\cdot|x) - p(\cdot|x)\|_1^2}{\min_{y \in \mathcal{Y}} p(y|x)} \right) = \log \left( 1 + \frac{2\alpha^2}{1 - 0.5\alpha} \right) \leq \log \left( 1 + 4\alpha^2 \right),$$

where we use the fact that $\alpha \leq 1$.

Setting $\alpha = \frac{1}{2} \sqrt{e^\epsilon - 1}$, for any $\epsilon \in [0, \log(5)]$, we have $\mathbb{E}_x \left[ \mathbb{D}_{\mathrm{KL}} \left( \mathbb{P}(\cdot|x), p(\cdot|x) \right) \right] \leq \epsilon$, and:

$$\mathbb{E}_x \left[ \left( \mathbb{E}_{y^* \sim \mathbb{P}(\cdot|x)} [y^*] - \hat{y}_{\mathrm{RAIL}}(x) \right)^2 \right] = \left( \frac{N}{4} \right)^2 (e^\epsilon - 1) \geq \left( \frac{N}{4} \right)^2 (1 + \epsilon - 1) = \left( \frac{N}{4} \right)^2 \epsilon,$$

where we use the fact that $e^\epsilon \geq 1 + \epsilon$. $\square$

## A.3 PROOF FOR LEMMA 3

*Proof.* For simplicity, we avoid explicitly stating conversions from float to string, and vice versa. For any $x$, we wish to minimize:

$$\mathbb{E}_{y^* \sim \mathbb{P}(\cdot|x)} \left[ \left( \sum_{y \in \mathcal{Y}_{\mathrm{grid}}} p(y|x) \cdot y - y^* \right)^2 \right].$$

Equating the derivative w.r.t. $p(y|x)$ to 0, we derive the first-order condition for optimality:

$$2 \cdot \mathbb{E}_{y^* \sim \mathbb{P}(\cdot|x)} \left[ \left( \sum_{y' \in \mathcal{Y}_{\text{grid}}} p(y'|x) \cdot y' - y^* \right) \right] \cdot y = 0, \forall y \in \mathcal{Y}_{\text{grid}}.$$

So the optimal solution is achieved when:

$$\sum_{y' \in \mathcal{Y}_{\text{grid}}} p(y'|x) \cdot y' = \mathbb{E}_{y^* \sim \mathbb{P}(\cdot|x)}[y^*].$$

From the conditions in the lemma, we have that the smallest and largest numbers in $\mathcal{Y}$ are present in $\mathcal{Y}_{\text{grid}}$. Since $\mathbb{E}_{y^* \sim \mathbb{P}(\cdot|x)}[y^*]$ can be expressed as a convex combination of the smallest and largest numbers in $\mathcal{Y}$, it can also be expressed as a convex combination of numbers in $\mathcal{Y}_{\text{grid}}$. Hence, there exists a probability distribution $p^*(y|x)$ such that $\hat{y}_{\text{RAIL}}(x; \mathcal{Y}_{\text{grid}}) = \sum_{y' \in \mathcal{Y}_{\text{grid}}} p^*(y' \mid x) \cdot y' = \mathbb{E}_{y^* \sim \mathbb{P}(\cdot|x)}[y^*]$, thus satisfying the condition for optimality. $\qquad \square$

# B RELATED WORKS

*Encoder-based* models (e.g., BERT) relying on the masked language modeling pretraining tasks have been primarily employed to *discriminative* tasks (including classification and regression) (Devlin et al., 2019). *Decoder-based* large language models (LLMs) (e.g., GPT, LLaMa) on the other hand, mostly relying on the next-token prediction pretraining task, showed state of the art results across a range of generative tasks. (OpenAI et al., 2023; Anil et al., 2023; Touvron et al., 2023; Gemini Team et al., 2024) While there is an on-going research regarding whether the encoder or decoder architecture is better tailored to predictive tasks (Nityasya et al., 2023; Li et al., 2023; Dukić & Snajder, 2024; Qorib et al., 2024), in this work we focus on the question of *how do we best apply decoder models to predictive tasks?*

Generative models have been successfuly applied to numeric prediction, where a number is generated autoregressively token-by-token. For example, Gruver et al. (2023); Requeima et al. (2024) considered it in a zero-shot learning setup for time series prediction, Vacareanu et al. (2024a) experimented with zero-shot regression problems, and Liu & Low (2023); Yang et al. (2023) considered the autoregressive finetuning over numerical targets applied to arithmetic tasks. Multiple prior works used predictive distribution from an LLM towards improving the prediction, including the median rule (Gruver et al., 2023; Requeima et al., 2024) and more broadly, Bayes-optimal rule optimizing a regression metric of choice (Lukasik et al., 2024). The importance of tokenizing the numerical targets into individual digits has been raised by previous works (Liu & Low, 2023; Yang et al., 2023), and importance of prompt selection was analyzed by Requeima et al. (2024).

## C    COMPARISON TO MBR-BASED FINE-TUNING

The RAFT loss may be contrast against ideas in the literature on Minimum Bayes Risk (MBR) prediction literature (Kaiser et al., 2000; Shannon, 2017; Prabhavalkar et al., 2018), which optimize *non-regression metrics* via approximation using sampled model predictions. For the squared loss, this may be formulated as follows:

**Definition 2.**  Define the *sampled regression-aware loss* as follows:

$$\ell_{\text{MBR}}(y^*, p(\cdot \mid x)) = \mathbb{E}_{y \sim p(\cdot \mid x)}\left[ (y^* - \texttt{float}(y))^2 \right]. \tag{13}$$

Compared to the loss in Equation 9, the key difference is that the expectation over the model outputs appears *outside* the square loss. A naïve empirical implementation of this objective requires explicitly sample responses from the model $p(\cdot \mid x)$; this can be expensive and incur high variance. As with $\hat{y}_{\text{RAIL}}(x; \mathcal{Y}_{\text{grid}})$, one may instead consider a practical variant that approximates the expectation using a restricted grid of targets $\mathcal{Y}_{\text{grid}} \subset \mathcal{Y}$:

$$\hat{L}_{\text{MBR}}(p; \mathcal{Y}_{\text{grid}}) = \frac{1}{N} \sum_{(x, y^*) \in S} \sum_{y \in \mathcal{Y}_{\text{grid}}} p(\texttt{str}(y) \mid x) \cdot (y^* - y)^2 \tag{14}$$

Even this variant has a notable disadvantage: the minimizer of Equation 14 is a one-hot distribution that places all its probability mass on one of the targets in $\mathcal{Y}_{\text{grid}} \subset \mathcal{Y}$:

**Lemma 4.**  Let $y^*(x) = \mathbb{E}_{y^* \sim \mathbb{P}(\cdot \mid x)}[y^*]$ denote the Bayes-optimal prediction for input $x$. We assume $\mathbb{P}(\cdot \mid x)$ is supported on numerical targets only. The minimizer of the approximate *sampled regression-aware* loss in Equation 14 over all model distributions $p(\cdot \mid x)$ is of the form:

$$p(y \mid x) = \begin{cases} 1 & \text{if } y = \arg\min_{y' \in \mathcal{Y}_{\text{grid}}} \|y' - y^*(x)\|_2 \\ 0 & \text{else} \end{cases}.$$

Therefore, the quality of the minimizer $p(\cdot \mid x)$ entirely depends on how well $\mathcal{Y}_{\text{grid}}$ approximates the original target space $\mathcal{Y}$. For example, if $\mathcal{Y}_{\text{grid}}$ is a set of integers, the minimizer of Equation 14 will also be limited to predicting integers, even when the original target space $\mathcal{Y}$ contains floating-point numbers of arbitrary precision. As shown in Lemma 3, RAFT does not suffer from the loss of precision resulting from the use an approximate target space, and also avoids the high variance associated with sampling. In Table 5, we experimentally verify better performance of RAFT over the MBR-based fine-tuning, and unless otherwise stated, we focus our attention to RAFT. Also, see §E.8 for an analysis of the RAFT predictor in terms of the learnt distributions over tokens.

We next prove Lemma 4.

*Proof.*  Notice that:

$$\mathbb{E}_{y^* \sim \mathbb{P}(\cdot \mid x)}\left[ \mathbb{E}_{y \sim p(\cdot \mid x)}\left[ (y - y^*)^2 \right] \right] = \mathbb{E}_{y^* \sim \mathbb{P}(\cdot \mid x)}\left[ \sum_y p(y \mid x) \cdot (y - y^*)^2 \right]$$

$$= \sum_y p(y \mid x) \cdot \mathbb{E}_{y^* \sim \mathbb{P}(\cdot \mid x)}\left[ (y - y^*)^2 \right].$$

Since this is a convex combination, the optimal value is achieved for $p$ to be a one-hot vector with a 1 on the index $\arg\min_y \mathbb{E}_{y^* \sim \mathbb{P}(\cdot \mid x)}\left[ (y - y^*)^2 \right]$. We thus want a $y$ that minimizes:

$$\mathbb{E}_{y^* \sim \mathbb{P}(\cdot \mid x)}\left[ (y - y^*)^2 \right] = \mathbb{E}_{y^* \sim \mathbb{P}(\cdot \mid x)}\left[ y^2 + (y^*)^2 - 2 \cdot y \cdot y^* \right].$$

Equivalently, we want a $y$ that minimizes:

$$y^2 - 2 \cdot y \cdot \mathbb{E}_{y^* \sim \mathbb{P}(\cdot \mid x)}[y^*].$$

or equivalently:

$$y^2 - 2 \cdot y \cdot \mathbb{E}_{y^* \sim \mathbb{P}(\cdot \mid x)}[y^*] + (\mathbb{E}_{y^* \sim \mathbb{P}(\cdot \mid x)}[y^*])^2 = (y - \mathbb{E}_{y^* \sim \mathbb{P}(\cdot \mid x)}[y^*])^2.$$

$\square$

| Method/ablation | RMSE |
|---|---|
| Sampled regression aware (Definition (2)) | 0.98 |
| Regression aware (Definition (1)) | **0.40** |

Table 5: Root mean squared error (RMSE) on STSB across regression aware approaches and their variants on Gemma-2 9B.

We next provide a general version of Lemma 4.

**Lemma 5.** The minimizer of the following objective:

$$\mathbb{E}_{y^* \sim p^*(\cdot|x)} \mathbb{E}_{y \sim p(\cdot|x)} \left[ \ell(y^*, y) \right],$$

for a loss function $\ell : \mathcal{Y} \times \mathcal{Y} \to \mathbb{R}_+$, is a one-hot distribution over targets such that all probability mass is on a target $\hat{y} \in \mathcal{Y}$ which minimizes $\mathbb{E}_{y^* \sim p^*(\cdot|x)} \left[ \ell(y^*, \hat{y}) \right]$.

*Proof.* The proof is elementary. Expanding the above objective:

$$
\begin{aligned}
\mathbb{E}_{y^* \sim p^*(\cdot|x)} \mathbb{E}_{y \sim p(\cdot|x)} \left[ \ell(y^*, y) \right] &= \mathbb{E}_{y \sim p(\cdot|x)} \mathbb{E}_{y^* \sim p^*(\cdot|x)} \left[ \ell(y^*, y) \right] \\
&= \int_{y \in \mathcal{Y}} \mathbb{E}_{y^* \sim p^*(\cdot|x)} \left[ \ell(y^*, y) \right] \cdot p(y|x) \cdot \mathrm{d}y \\
&\geq \int_{y \in \mathcal{Y}} \mathbb{E}_{y^* \sim p^*(\cdot|x)} \left[ \ell(y^*, \hat{y}) \right] \cdot p(y|x) \cdot \mathrm{d}y \\
&= \mathbb{E}_{y^* \sim p^*(\cdot|x)} \left[ \ell(y^*, \hat{y}) \right] \\
&= \int_{y \in \mathcal{Y}} \mathbb{E}_{y^* \sim p^*(\cdot|x)} \left[ \ell(y^*, y) \right] \cdot p(y|x) \cdot \mathrm{d}y,
\end{aligned}
$$

where the third step follows from the fact that $\hat{y}$ minimizes $\mathbb{E}_{y^* \sim p^*(\cdot|x)} \left[ \ell(y^*, \cdot) \right]$; on the final step, $p(\cdot|x)$ is a probability distribution that has a point mass on $\hat{y}$. $\square$

# D    ADDITIONAL EXPERIMENTAL DETAILS

We update all parameters during the fine-tuning. We summarize specific settings below. For Gemma-2, we use the following settings:

- We use dropout rate $0.1$ and batch size $16$.
- We train for $200K$ steps and select the best step using the held out validation set (see Table 7 for details on the train/test/validation splits).
- We use a constant learning rate schedule. We select the learning rate value over the validation set from the values: $10^{-4}, 10^{-5}, 10^{-6}$.
- We use the Adafactor optimizer to save memory during the fine-tuning (we find Adam to not perform better). The parameters for Adafactor are: $\epsilon_1 = 10^{-30}$, $\epsilon_2 = 10^{-3}$, decay rate $= 0.8$.

For PaLM-2, we use the above settings, except we use batch size $64$ and no dropout, and train for $5K$ steps and report the results from the last checkpoint.

For MovieLens, we use AdamW optimizer and sweep learning rates from the range $\{10^{-4}, 10^{-5}, 10^{-6}\}$. We use a cosine decay schedule for the learning rate, with $10K$ steps of warmup from learning rate $10^{-8}$ .

In Table 6, we report the prompts we used in our experiments, and in Table 7 we report the dataset statistics.

| Dataset | Input prompt | Target range |
|---|---|---|
| STSB | What is the sentence similarity between the following two sentences measured on a scale of 0 to 5: {Sentence #1}, {Sentence #2}. The similarity measured on a scale of 0 to 5 with 0 being unrelated and 5 being related is equal to | [0, 5] |
| Amazon reviews | product_category: {product category} product_title: {product title} review_date: {review date} review_headline: {review headline} review_body: {review body} Question: In a star rating of 1, 2, 3, 4, 5, the higher the better, what would be the star rating of the above review? Please only give me the final rating and I do not need any explanations. | 1, 2, 3, 4, 5 |
| MovieLens-1M | Instruction: Predict the rating of a target movie based on the user's historical movie ratings. Rating History: {Rating history} Candidate Item: {Candidate Item}. Output: | 1, 2, 3, 4, 5 |
| Synthetic (Original #1 from (Vacareanu et al., 2024a)) | The task is to provide your best estimate for 'output score' based on 'input score'. Please provide that and only that, without any additional text. Input score: {Input score}. Output score: | [0, 9] |

Table 6: Prompts used for different datasets and the corresponding target ranges. Curly braces denote inputs specific to an input example. For Synthetic (Original #1 from (Vacareanu et al., 2024a)) we normalize the targets to correspond to [0, 9].

| Dataset | Train size | Validation size | test size |
|---|---|---|---|
| Wireless | 10,000 | 1,500 | 1,500 |
| Personal care | 10,000 | 1,500 | 1,500 |
| Music | 10,000 | 1,500 | 1,500 |
| STSB | 4,887 | 863 | 1,500 |
| STSB 1k | 1,000 | 863 | 1,500 |
| MovieLens-1M | 797,758 | 10,145 | 10,145 |
| Synthetic (Original #1 from (Vacareanu et al., 2024a)) | 10,000 | 1,000 | 1,000 |

Table 7: Summary of dataset statistics.

# E    ADDITIONAL EXPERIMENTAL RESULTS

We provide additional experimental results corroborating the findings in the main paper.

## E.1    STSB-1$K$

In Table 8, we report results from sub-sampling the training set for STSB down to $1K$ examples. We find similar trends as observed in the case of other datasets in Table 3.

| Model size | Zero-shot standard decoding | Standard fine-tuning standard decoding | Predictive head | RAFT |
|---|---|---|---|---|
| 2B | 1.10 | 0.61±0.01 | **0.58±0.01** | **0.58±0.00** |
| 9B | 1.31 | 0.57±0.01 | 0.57±0.01 | **0.56±0.01** |

Table 8: RMSE on STSB-1$K$ across methods, and Gemma-2 models of varying sizes. Fine-tuning methods report mean $\pm$ std dev from model retraining.

## E.2    SYNTHETIC DATA

For a simple setting, we consider a synthetic regression dataset from Vacareanu et al. (2024a) referred to as *the Original #1 dataset* by the authors. We report results in Table 9 and corroborate our observation from the language regression task experiment that RAFT improves over the predictive head approach when initialized from a pre-trained checkpoint, and not when the model weights are initialized randomly. This provides additional support for our hypothesis that the alignment of RAFT to the next-token prediction pre-training task is the underlying reason for its better performance over the predictive head.

| initialiation | model size | autoregressive | predictive head | RAFT |
|---|---|---|---|---|
| Pre-trained checkpoint | 2B | 0.018 | 0.013 | **0.005** |
|  | 9B | 0.017 | 0.017 | **0.006** |
| Random | 2B | 2.536 | **0.327** | 1.704 |
|  | 9B | 2.536 | **0.147** | 1.092 |

Table 9: The role of initialization to the pre-trained checkpoint on a synthetic regression dataset from Vacareanu et al. (2024a) (the *Original #1* dataset). We compare RMSE across different Gemma model sizes, and across different fine-tuning methods: *autoregressive*, *predictive head* and *autoregressive regression aware*. We corroborate our observation from language regression tasks that RAFT improves over the predictive head approach when initialized from a pre-trained checkpoint, and not when model weights are initialized randomly.

## E.3    GEMMA-2 27B AND PALM-2 MODELS

We report results on Gemma-2 27B across all dataset in Table 10 and on PALM-2 models on STSB in Table 11, corroborating the findings of RAFT improving in most settings.

## E.4    COMPARISON TO ENCODER-BASED BASELINES

In Table 12, we report results with additional baselines that use a prediction head over RoBERTa representation for the input sequence. We include: mean-pooling and CLS token variants, and both frozen RoBERTa weights and unfrozen weights in the fine-tuning. We also include the SMART method (Jiang et al., 2019). In keeping with previous work, in addition to RMSE, we also report performance on the Pearson and Spearman metrics for STSB (Jiang et al., 2019) (where available). We find RAFT to surpass all the included baselines.

| dataset | zero-shot | autoregressive | predictive head | RAFT |
|---|---|---|---|---|
| Wireless | 0.87 | 0.83 | 0.50 | **0.44** |
| Personal care | 0.94 | 0.89 | 0.50 | **0.47** |
| Music | 1.26 | 1.23 | **0.47** | 0.49 |
| STSB 1k | 1.29 | 0.60 | 0.56 | **0.55** |
| STSB | 1.29 | 0.62 | 0.51 | **0.48** |

Table 10: Comparison of RMSE across datasets for Gemma-2 27B. In most cases, RAFT outperforms all other methods.

| model family | model size | autoregressive | predictive head | RAFT |
|---|---|---|---|---|
| PALM-2 | 1B | $0.79 \pm 0.02$ | $\mathbf{0.61} \pm 0.01$ | $0.62 \pm 0.01$ |
| PALM-2 | 24B | $0.63 \pm 0.03$ | $0.56 \pm 0.00$ | $\mathbf{0.53} \pm 0.00$ |

Table 11: Comparison of root mean squared error (RMSE) on STSB across different PALM-2 model sizes, and across different fine-tuning methods: *autoregressive*, *predictive head* and *autoregressive regression aware*. Each model is ran for 3 times to obtain standard deviations. For PALM-2 1B, we find no difference in the performance of predictive head and RAFT, and both outperform the autoregressive approach. For PALM-2 24B, we see a significant improvement from RAFT over both the autoregressive and the predictive head approaches.

### E.5 CHOICES FOR TOKENS IN RAFT

**Sensitivity to the grid size**     Following Lemma 3, $\hat{y}_{\mathrm{RAIL}}(x)$ can express any numerical target in the population limit, even with a coarse grid. Empirically, however, one might expect different grids to affect the results. We now assess this point on the Amazon reviews Wireless dataset, comparing the following choices for $\mathcal{Y}_{\mathrm{grid}}$:

(1) { '1', '2', '3', '4', '5' } (the default choice for RAFT),

(2) { '5' } (*viz.* $\max(\mathcal{Y})$),

(3) { '1','9' } (the smallest and largest digit in $\mathcal{V}$),

(4) { '4', '5' } (the two largest digits in $\mathcal{Y}$ for all datasets),

(5) { '7','8','9' } (the two largest digits in $\mathcal{V}$).

| Method/ablation | Parameter count | RMSE | Pearson corr. | Spearman corr. |
|---|---|---|---|---|
| RoBERTa Base CLS | 110M | 0.64 | 90.84 | 90.59 |
| RoBERTa Large CLS | 356M | 0.59 | 91.99 | 92.02 |
| RoBERTa Large mean-pooling | 356M | 0.63 | 91.65 | 91.56 |
| RoBERTa Large mean-pooling freeze | 356M | 1.08 | 72.72 | 74.16 |
| RoBERTa Large CLS freeze | 356M | 1.30 | 56.48 | 54.76 |
| SMART BERT (Jiang et al., 2019) | 356M | - | 90.00 | 89.40 |
| SMART RoBERTa (Jiang et al., 2019) | 356M | - | 92.80 | 92.60 |
| Gemma-2 2B RAFT | 2B | 0.54 | 93.55 | 93.22 |
| Gemma-2 9B RAFT | 9B | **0.51** | **94.30** | **94.18** |

Table 12: Parameter counts and performance metrics on STSB across baselines. Results from SMART (Jiang et al., 2019) taken as reported in the paper (RMSE was not reported).

| Dataset | Model size | '1'-'5' | '5' | '1','9' | '4','5' | '7','8','9' |
|---|---|---|---|---|---|---|
| Wireless | 2B | **0.47±0.01** | 0.48±0.01 | 0.48±0.00 | 0.48±0.00 | 0.48±0.01 |
|  | 9B | **0.45±0.01** | **0.45±0.01** | 0.47±0.02 | 0.46±0.00 | 0.47±0.01 |
| Personal care | 2B | **0.49±0.00** | **0.49±0.00** | **0.49±0.02** | **0.49±0.00** | **0.49±0.00** |
|  | 9B | **0.47±0.01** | 0.48±0.01 | 0.48±0.02 | **0.47±0.00** | **0.47±0.00** |
| Music | 2B | **0.50±0.00** | **0.50±0.00** | **0.50±0.01** | **0.50±0.00** | **0.50±0.01** |
|  | 9B | **0.46±0.01** | 0.47±0.00 | 0.48±0.02 | 0.47±0.00 | 0.48±0.00 |

Table 13: Comparison of RMSE (mean ± std dev) across variants of RAFT with varying sizes of the grid $\mathcal{Y}_{\text{grid}}$. The choice of '1'-'5' outperforms the alternatives.

In Table 13, we report the results for different choices for $\mathcal{Y}_{\text{grid}}$. We find that, aligned with Lemma 3, limiting the grid does not significantly impact the results. However, when comparing the number of steps to convergence (see Table 21), we find that the default choice (1) in most cases tends to converge faster to the best solution than other choices. One explanation for this finding is the following. As shown by Lukasik et al. (2024), the choice of { '1', '2', '3', '4', '5' } yields a reasonable result for the zero-shot RAIL approach, and thus it provides a good starting point for fine-tuning. Recall that the zero-shot RAIL corresponds to the RAFT approach at step 0 of training, since the predictors for each are equivalent.

**Sensitivity to the grid token indices**   The next question we pose is about the importance of strictly sticking to numeric tokens: what would happen if the RAFT predictor $\hat{y}_{\text{RAFT}}(x)$ used non-numeric tokens? To analyze this question, let us consider a more general form of the predictor:

$$\hat{y}_{\text{RAFT-NN}}(x) = \sum_{y \in \mathcal{Y}_{\text{grid}}} p(\texttt{token}(y) \mid x) \cdot y, \tag{15}$$

where $\texttt{token}(y) \in \mathcal{V}$ denotes a token of choice corresponding to the numerical target $y$.

We keep $\mathcal{Y}_{\text{grid}}$ as composed of the digits { '1', '2', '3', '4', '5' }, and use the predictor in Equation 15 with the following choices for tokens:

(1) token for each digit becomes an alphabet token starting with 'a' and ending with 'a',

(2) each token is a digit (i.e., '1' becomes '5', '2' becomes '4'),

(3) we only consider digit '5'.

As shown in Table 14, in most settings, the choice of digits { '1', '2', '3', '4', '5' } is as performant as any other choice. In certain settings, we even find a large drop in performance (e.g., the choice of characters or months). However, by enlarge, we find the results do not worsen significantly when different tokens are used.

E.6   CHOICES FOR LOSS, NORMALIZATION AND TOKEN INITIALIZATION IN RAFT

In Table 15 we compare the MSE loss to distillation style log loss on STSB. The target is scaled to be between 0 and 1 for this set of experiments. The log loss is defined as $-y^* \log p_1 - (1-y^*) \log(1-p_1)$ where $p_1$ is the probability of digit '1'. We find that both the MSE loss and the log loss yield similar results, and that scaling the range of the targets does not negatively affect the performance.

We next analyze whether enforcing the probabilities over the grid to sum to 1 (by normalizing them by the sum of the probabilities of all numbers in $\mathcal{Y}_{\text{grid}}$) can improve the performance of RAFT. Table 16 shows that applying normalization to the probabilities for numbers in $\mathcal{Y}_{\text{grid}}$ does not significantly affect the results.

In Table 17, we compare three initialization methods for the embedding of numbers in autoregressive and RAFT methods. The tokens used are of granularity of 0.1 for both autoregressive and RAFT methods, except for RAFT '1'-'5', where digits from 1 to 5 are used for the grid (granularity 1.0).

| dataset | model size | digits '1'–'5' | characters 'a'-'e' | reversed digits | 'January'-'May' |
|---|---|---|---|---|---|
| Wireless | 2B | $0.47 \pm 0.01$ | $0.48 \pm 0.01$ | $0.48 \pm 0.00$ | $0.83 \pm 0.01$ |
| | 9B | $0.45 \pm 0.00$ | $0.46 \pm 0.00$ | $0.46 \pm 0.01$ | $0.47 \pm 0.01$ |
| Personal care | 2B | $0.49 \pm 0.00$ | $0.48 \pm 0.00$ | $0.48 \pm 0.01$ | $0.85 \pm 0.00$ |
| | 9B | $0.47 \pm 0.01$ | $0.47 \pm 0.00$ | $0.48 \pm 0.01$ | $0.48 \pm 0.01$ |
| Music | 2B | $0.50 \pm 0.00$ | $0.50 \pm 0.01$ | $0.50 \pm 0.01$ | $0.50 \pm 0.00$ |
| | 9B | $0.46 \pm 0.00$ | $0.48 \pm 0.01$ | $0.47 \pm 0.01$ | $0.61 \pm 0.25$ |

Table 14: Comparison of RMSE (mean $\pm$ std dev) across variants of RAFT where different tokens are used for the prediction formula of RAFT. Each experiment repeated for 3 times. In most settings, the choice of digits '1'–'5' is at least as performant as any other choice. In certain settings, we find a large drop in performance compared to the choice of digits (i.e., months 'January'-'May'.

| | unscaled targets | | targets scaled to $[0, 1]$ | |
|---|---|---|---|---|
| model size | predictive head | RAFT | RAFT '0' and '1' | log loss |
| 1B | $0.61 \pm 0.01$ | $0.62 \pm 0.01$ | $0.63 \pm 0.01$ | $0.63 \pm 0.01$ |
| 24B | $0.56 \pm 0.00$ | $0.53 \pm 0.00$ | $0.54 \pm 0.01$ | $0.53 \pm 0.00$ |

Table 15: Comparision of RMSE (mean $\pm$ std dev) on the STSB dataset for MSE loss and log loss when the target is scaled to be between 0 and 1. The models are PALM-2 1B and 24B models. The MSE loss uses digit '0' and '1' to compute the predicted value. The log loss is in the form $-y^* \log p_1 - (1 - y^*) \log(1 - p_1)$ where $p_1$ is the probability of digit '1'.

Overall, we find that RAFT is less sensitive to the initialization method than the autoregressive approach.

In Table 18, we experiment with random initialization for the tokens in the RAFT grid. We find random initialization to lead to worse results compared to using pre-trained token embeddings.

We next analyze the impact of the choice of $\mathcal{Y}_{\text{grid}}$ in computing the RAIL predictor. To this end, we vary the granularity of $\mathcal{Y}_{\text{grid}}$ by constructing a list of equally spaced numbers covering the range of $\mathcal{Y}$. For example, choosing granularity to be 0.1 when $\mathcal{Y} = [0, 5]$ yields $\mathcal{Y}_{\text{grid}} = \{$'0.0', '0.1', '0.2', $\ldots$, '4.9', '5.0' $\}$. In our implementation, all numbers in $\mathcal{Y}_{\text{grid}}$ are represented by single tokens that we add to the vocabulary. We initialize the token embedding with either the *First* or the *Average* method. In the *First* method, we initialize the embedding with the embedding of the first digit of the number (e.g. use token '0' embedding for the number '0.1'). In the *Average* method, we initialize the embedding with the average of the embedding from the constituent tokens (e.g. use the average embeding of token '0', '.' and '1' for the number '0.1'.) We report the results on the STSB datasets with PALM-2 1B model in Table 19 and find no significant difference in the results across different choices for the granularity of $\mathcal{Y}_{\text{grid}}$.

Additionally, in Table 19 we also include the autoregressive method utilizing additional tokens from $\mathcal{Y}_{\text{grid}}$ as constructed with the methodology outlined above (i.e., with varying granularity). Here, contrary to RAFT, we find the initialization method to affect the results, with *First* performing better than *Average*. Note that the autoregressive method is equivalent to the generative classification from (Fernandes et al., 2023), where the classes correspond to the numbers from the grid.

Lastly, we would like to note that in the case of generative classification, there is a trade-off between how fine-grained the grid is and how many examples per token are observed during training. In particular, if the classes are too coarse, we observe a loss in performance. On the other hand, if the classes are too fine-grained, there may be insufficient training examples per label to learn the embeddings for new tokens. For example, with granularity 0.05, 17 out of the 101 numbers in the grid do not appear in the training data. This can also lead to a loss in performance.

| model size | not normalized | normalized |
|---|---|---|
| 1B | $0.63 \pm 0.00$ | $0.63 \pm 0.02$ |
| 24B | $0.53 \pm 0.00$ | $0.53 \pm 0.01$ |

Table 16: The effect of normalization of grid token probabilities in RAFT on the STSB dataset. The models are PALM-2 1B and 24B models.

| initialization | autoregressive | RAFT | RAFT '1'-'5' |
|---|---|---|---|
| Zero | $1.20 \pm 0.01$ | $0.62 \pm 0.00$ | $0.63 \pm 0.00$ |
| First | $0.80 \pm 0.02$ | $0.63 \pm 0.00$ | $0.62 \pm 0.02$ |
| Average | $0.98 \pm 0.01$ | $0.64 \pm 0.01$ | $0.61 \pm 0.01$ |

Table 17: Comparison of different initialization methods for embeddings for tokens corresponding to numbers in autoregressive and RAFT methods on STSB dataset with PALM-2 1B model. We consider granularity 0.1 for constructing the tokens for autoregressive and RAFT reported in the first 2 columns, whereas the final column reports RAFT with tokens 1 to 5. *Zero* denotes initialization with 0 values, *First* denotes the initialization with the embedding corresponding to the first token of the number (e.g. use token '0' embedding for the number '0.1'), and *Average* denotes the initialization with the embedding corresponding to the different tokens in the number (e.g. average embeddings for tokens '0', '.' and '1' when initializing the embedding for the number '0.1'). RAFT is less sensitive to the initialization method. We find the performance does not significantly worsen with different initialization methods compared to using the pre-trained token embeddings at initialization (see Table 15).

| RAFT with '1'-'5' | | RAFT with '5' | |
|---|---|---|---|
| baseline | random | baseline | random |
| $0.63 \pm 0.00$ | $0.66 \pm 0.02$ | $0.62 \pm 0.01$ | $0.64 \pm 0.01$ |

Table 18: RMSE across different initialization (baseline and random) of the tokens in RAFT. We find random initialization of tokens in the RAFT grid to hurt the performance. The model is the PALM-2 1B model.

| | autoregressive | | RAFT '1'-'5' | |
|---|---|---|---|---|
| granularity | First | Average | First | Average |
| 0.05 | $0.85 \pm 0.01$ | $1.11 \pm 0.01$ | $0.61 \pm 0.01$ | $0.63 \pm 0.00$ |
| 0.1 | $0.80 \pm 0.02$ | $0.98 \pm 0.01$ | $0.63 \pm 0.00$ | $0.64 \pm 0.01$ |
| 0.2 | $0.83 \pm 0.02$ | $0.89 \pm 0.01$ | $0.64 \pm 0.01$ | $0.66 \pm 0.02$ |
| 0.5 | $0.84 \pm 0.01$ | $0.87 \pm 0.01$ | $0.63 \pm 0.01$ | $0.63 \pm 0.02$ |
| 1.0 | $0.81 \pm 0.01$ | $0.85 \pm 0.01$ | $0.62 \pm 0.02$ | $0.61 \pm 0.01$ |

Table 19: Comparison of RMSE on autoregressive and RAFT methods with different granularity of tokens. We append tokens that represent numbers with various granularity. For example, with granularity 0.05, the tokens are '0.00, '0.05, '0.10', ..., '5.00'. For autoregressive, this is equivalent to generative classification method from (Fernandes et al., 2023). *First* and *Average* denotes different ways to initialize the embeddings of the tokens, with details described in the caption of Table 17. The model is the PALM-2 1B model.

| model size | '1'−'5' | '5' | '1'−'9' | 'b'-'f' | '5'−'1' | ''Jan.'-'May'' |
|---|---|---|---|---|---|---|
| 1B | 0.63 ± 0.00 | 0.62 ± 0.01 | 0.63 ± 0.01 | 0.61 ± 0.01 | 0.62 ± 0.01 | 0.64 ± 0.01 |
| 24B | 0.53 ± 0.00 | 0.53 ± 0.01 | 0.53 ± 0.00 | 0.54 ± 0.02 | 0.54 ± 0.00 | 0.55 ± 0.01 |

Table 20: Comparison of RMSE (mean ± std dev) on the STSB dataset across variants of RAFT where different tokens are used to construct the grid for computing the RAFT predictor. The models are the PALM-2 24B and 1B models.

| dataset | model size | '1'-'5' | '5' | '1','9' | '4','5' | '7','8','9' |
|---|---|---|---|---|---|---|
| Wireless | 2B | **1,000** | 3,000 | 2,000 | 1,600 | 2,200 |
| | 9B | **1,600** | 2,800 | 3,400 | 3,200 | 4,200 |
| Personal care | 2B | 2,200 | 2,200 | 2,400 | **1,600** | 2,200 |
| | 9B | 4,200 | 2,000 | 6,200 | **400** | 2,400 |
| Music | 2B | **800** | 1,200 | 1,800 | 1,800 | 1,800 |
| | 9B | **1,000** | 2,200 | 3,000 | 1,600 | 3,000 |

Table 21: Comparison of the number of steps to convergence of RAFT where different number of numerical targets are used for the grid in RAFT. Results on the Amazon Wireless dataset.

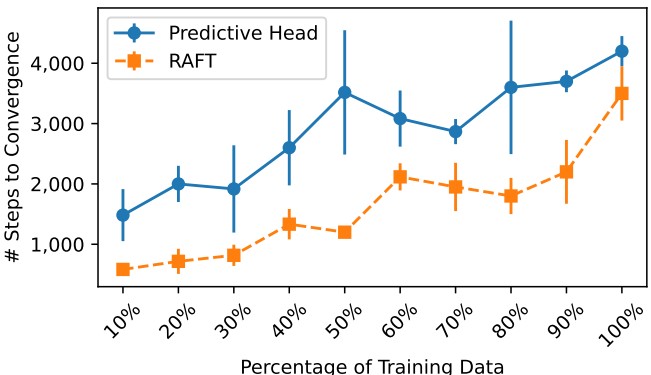

Figure 4: Comparison of the number of steps to convergence of RAFT and predictive head, with different percentage of the training set.

E.7 CONVERGENCE SPEED OF RAFT

In Table 21 we report the number of training steps to convergence to the best result on the held out validation set of different methods. We can see that RAFT with digits '1'-'5' it majority of cases converges faster than other choices for the grid.

In Figure 4 we compare the number of steps to convergence of RAFT and predictive head on STSB. The figure shows that RAFT converges with fewer number of steps across different percentages of training data used for training.

In Figure 5 we report RMSE on the test set as a function of the training step. We find that compared to predictive head, RAFT starts from a much lower RMSE score (as it corresponds to the RAIL method before training), and then converges in fewer steps.

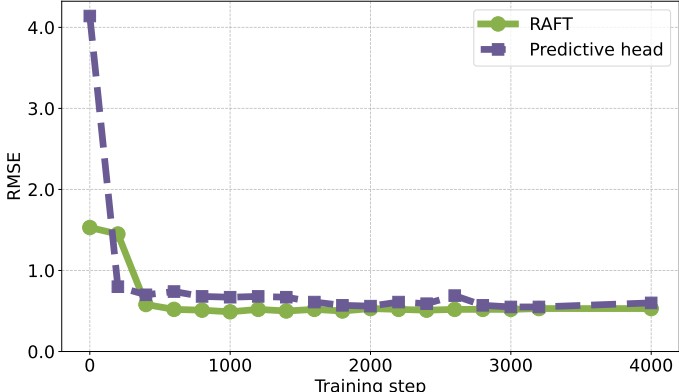

Figure 5: RMSE on the test set as a function of the training step on Amazon Wireless for Gemma-2 2B. Compared to predictive head, RAFT starts from a much lower RMSE at step 0, and then converges to its best RMSE in fewer steps.

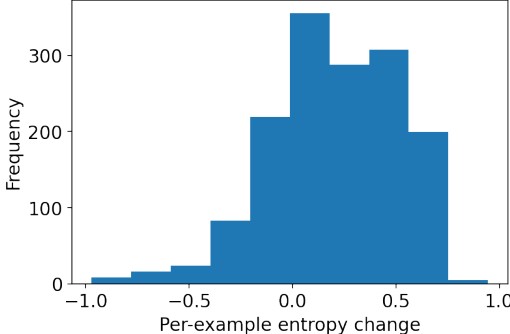

Figure 6: The histogram of per-example change (after vs. before fine-tuning) in entropy over the probabilities for the digits in the RAFT predictor. The entropy on average increases after the RAFT fine-tuning, meaning that the probabilities are overall more spread over the digits than prior to RAFT fine-tuning.

## E.8 DISTRIBUTION OVER TOKENS IN THE RAFT PREDICTOR

We next investigate the distribution over tokens in the RAFT predictor. We find that, while the error decreases with training, the entropy increases after RAFT training, as we show in Figure 6. This corresponds to the model on average spreading the probabilities over tokens more than prior to fine-tuning, as shown for specific examples in Figure 7. We posit this to be beneficial, and indeed, RAFT fine-tuning does not restrict uncertainty of the model, contrary to the MBR fine-tuning (compare Lemma 3 and Lemma 4).

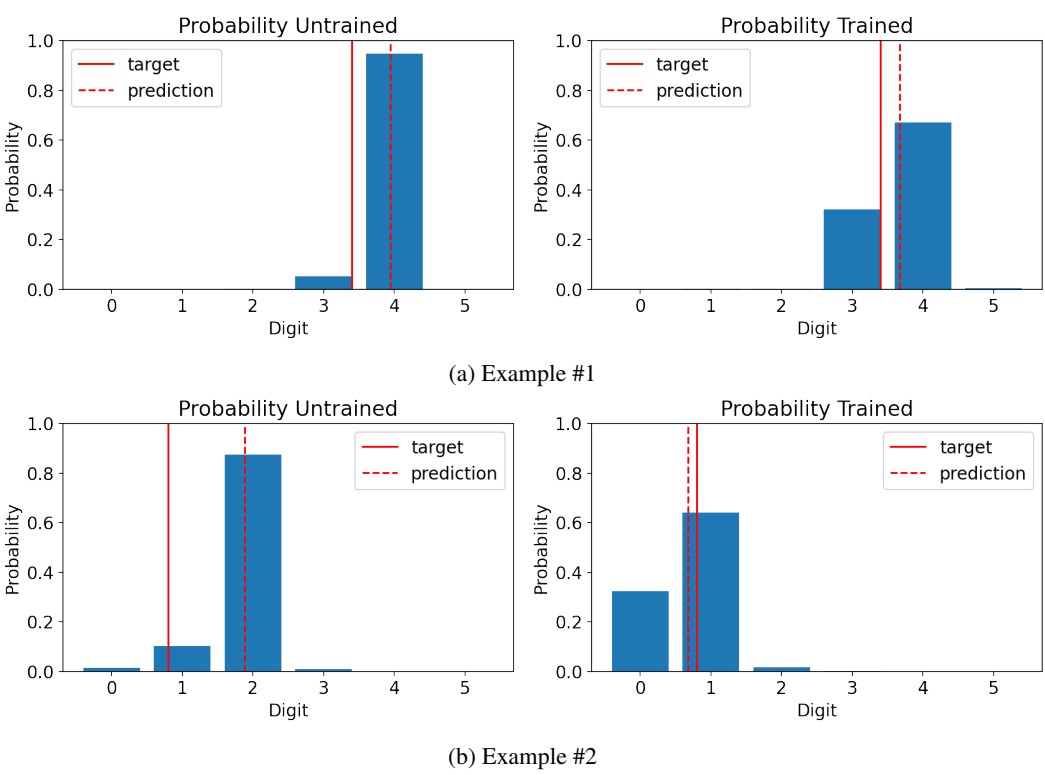

Figure 7: Comparison of the distribution over digit tokens in RAFT predictor (Gemma-2 2B on STSB) before and after fine-tuning. We find the entropy over the digit token probabilities overall increases after fine-tuning with the RAFT objective.

