# OpenReview forum: "Better autoregressive regression with LLMs via regression-aware fine-tuning"
_ICLR.cc/2025/Conference — ICLR 2025 Spotlight_

### Official Review · Reviewer_AkBL · 2024-10-29

**Soundness:** 3
**Presentation:** 3
**Contribution:** 1
**Rating:** 6
**Confidence:** 3

**Summary:**

This paper explores two approaches: (1) fine-tuning an LLM using the log-ppl loss with autoregressive sampling during inference, and (2) adding a predictive head and fine-tuning it with an appropriate loss function. Following an in-depth analysis, this paper introduces RAFT, a regression-aware, Bayes-optimal method for optimal LLM fine-tuning. Extensive experiments demonstrate that RAFT consistently outperforms baseline methods across various datasets and models.

**Strengths:**

1. This paper provides a well-rounded formulation for LLM-based regression approaches.
2. It presents thorough evaluations across diverse tasks (e.g., Wireless, Music, Personal Care) and models (e.g., Gemma and PaLM).
3. The proposed RAFT method demonstrates effectiveness across these benchmarks.

**Weaknesses:**

1. While this paper presents a strong approach, it addresses a relatively niche problem. In my view, the choice between autoregressive sampling and a predictive head largely depends on the specific task, making it less crucial to claim that your method is universally superior to both across all tasks.
2. The proposed method does not show a significant improvement over the predictive head. For instance, in Table 9, RAFT performs worse than the predictive head on the Music dataset.

**Questions:**

Could you clarify the scenarios in which a predictive head performs better versus those where autoregressive sampling is more effective? Also, is RAFT the optimal choice in all situations, or are there cases where other methods might be preferable?

---

> ### Author Response · Authors · 2024-11-22
> **Response to Reviewer AkBL**
>
> > While this paper presents a strong approach, it addresses a relatively niche problem.
>
> As we discuss in the introduction, we would like to point out that regression problems encompass a wide range of problems, including semantic similarity prediction (Cer et al., 2017), quality estimation (Kocmi & Federmann, 2023; Jain et al., 2023; Fernandes et al., 2023), and sentiment analysis (Zhang et al., 2024). Moreover, there has been a large interest in utilizing LLMs to regression problems, e.g. (Gruver et al., 2023; Liu & Low, 2023; Yang et al., 2023; Fernandes et al., 2023; Zhuang et al., 2023; Lukasik et al., 2024). Thus, we believe that it is a pertinent question as to what is the most effective approach to regression with LLMs.
>
>
> > The choice between autoregressive sampling and a predictive head largely depends on the specific task, making it less crucial to claim that your method is universally superior to both across all tasks.
>
> Note that our approach is more grounded theoretically than predictive head when utilizing autoregressively pre-trained LLMs. In other words, it is well motivated for decoder based LLMs across regression problems.
>
>
> > The proposed method does not show a significant improvement over the predictive head. For instance, in Table 9, RAFT performs worse than the predictive head on the Music dataset.
>
> Please see a common response to all reviewers “RAFT improvements over predictive head”.
>
> > Could you clarify the scenarios in which a predictive head performs better versus those where autoregressive sampling is more effective?
>
> Can we ask for a clarification of what the reviewer refers to as “autoregressive sampling”: RAFT or regular autoregressive fine-tuning + decoding? We will address either interpretation below by contrasting the following three scenarios: predictive head, autoregressive fine-tuning + decoding and RAFT.
>
> **Autoregressive fine-tuning + decoding** In our work, we prove autoregressive based approaches can be arbitrarily poor for any regression problem due to misalignment to the evaluation metric: see the newly added Lemma 1 expanding this theoretical understanding to autoregressive fine-tuning + vanilla decoding, and also the Lemma 2 treating autoregressive fine-tuning + MALI decoding.
>
> **Predictive head** The predictive head deviates from the pre-training task, making it unclear whether it is well suited for finetuning decoder-based models. We add additional synthetic experiments to expand the understanding of the role of pretraining for decoder-based model finetuning, and further corroborate how when using pretraining checkpoint from the next-token prediction task, even on a simple synthetic regression dataset, predictive head underperforms RAFT; please see the common response to all reviewers “RAFT improvements over predictive head”.
>
> **RAFT** We argue for several empirical and conceptual benefits of RAFT over predictive head. Please see “RAFT improvements over predictive head”.
>
>
> > Is RAFT the optimal choice in all situations, or are there cases where other methods might be preferable?
>
> Our work argues that RAFT is the most theoretically grounded approach. Moreover, at step zero it corresponds to MALI, and thus, provides a reasonable performance even at zero shot. It also converges much faster than predictive head approaches even where there is no significant difference in the final performance. (e.g. see Table 12 and Figure 1).

---

> > ### Author Response · Authors · 2024-11-26
> > **Response to Reviewer AkBL**
> >
> > Dear reviewer,
> >
> > we would like to check if there are any remaining concerns or questions regarding our paper that we can clarify.

---

> > > ### Comment · Reviewer_AkBL · 2024-11-26
> > >
> > > Dear Authors,
> > >
> > > Thanks for your detailed response. I will raise my score to 6.

---

### Official Review · Reviewer_CzSf · 2024-10-31

**Soundness:** 3
**Presentation:** 3
**Contribution:** 3
**Rating:** 8
**Confidence:** 3

**Summary:**

This work introduces a new fine-tuning method for autoregressive language models to improve their performance on regression tasks, such as sentiment analysis, semantic similarity, and quality estimation for translation.

This work argues that the cross-entropy loss is not suited to regression tasks, since small deviation of the model from the ground truth can lead to high regression error. Indeed, the cross-entropy objective does not take into account the numerical value associated with text tokens. Instead, authors propose a new fine-tuning method (RAFT) based on the "Metric-aware LLM inference for regression" (MALI) approach.

**Strengths:**

- The research question is relevant and interesting.
 - The presentation of MALI (previous work) is clear.
- The counter-example of Lemma 1 is good. Since it's short, I would include it in the main body if possible.
- The experiment on using semantically unrelated tokens for the regression task (Table 11) is interesting.

**Weaknesses:**

- On a first read, I misunderstood the implications of Lemma 1. I think it would be worth slightly rewriting the paragraph after Lemma 1. In particular, I would replace the sentence
> Intuitively, log-perplexity fine-tuning treats all “wrong” predictions the same, *and does not account for the magnitude of their differences from the ground-truth target*.

by

> Intuitively, log-perplexity fine-tuning treats all “wrong” predictions the same, *as it is unaware of difference in magnitude of the numerical values represented by the tokens. For example, assuming that "100" and "1000" are represented with a single token, placing $\epsilon$ too much mass on tokens representing "100" is penalized similarly as placing  $\epsilon$ too much mass on the token "1000".

If you have a better rephrasing, you can of course use something else, but I think the aforementioned sentence can be improved (e.g. using my example).

- On the line 371, you are explaining that you are experimenting with a reduced training set for the STSB benchmark (using 1k examples instead of the whole training set). I understand that it is interesting to vary the amount of training data, but why are you only reducing the training set size on STSB and not on the other datasets?

- **Main weakness**: While your paper studies regression tasks using autoregressive models, you observe on lines 465-467 that when training from scratch, the predictive head methods work best. Since encoder-only models such as BERT are widely used for tasks mentioned in the introduction (e.g. sentiment analysis, regression, ranking), it would be valuable to include a baseline of a large fine-tuned RoBERTa model on the same tasks that you present in the paper. Your findings would remain interesting and relevant in case the RoBERTa baseline was beating RAFT, but I think knowing how RAFT compares to fine-tuning RoBERTa is very relevant to practitioners. Therefore, I would suggest adding the following experiments (with the same hyperparameter tuning as RAFT):
    - Prediction head regression over a mean-pooling of RoBERTa representation (over sequence length)
    - Prediction head regression over the output for the CLS token of RoBERTa model
    - Both variants with either frozen RoBERTa weights, as well as unfrozen weights

- There could be more practical details on the fine-tuning process. Are you updating all the parameters of the model or did you freeze some of them (eg. update only the last linear layer)? Which optimizer are you using? If you are using Adam, what values of the betas, weight decay, and epsilon are you using? What batch size are you using? How long does one training run takes on average? How many machines did you need? The answers to these questions are important to help other researchers reproduce your results.

**Questions:**

- I don't understand why the authors are comparing MALI (regression on continuous values) against the Minimum Bayes Risk (MBR) prediction literature, since it optimizes non-regression metrics on the discrete grid (section 3.4). There is one small experiment comparing the sampled regression aware approach (table 8), but it is only in the appendix. It would be helpful to either strengthen the connection between MBR and MALI/RAFT, or consider moving this section to the appendix if it's not central to the paper's main argument. If you choose to keep it in the main text, consider explaining more clearly why this comparison is important for understanding RAFT's contributions.

- Similar to the experiment presented in Table 11, did the authors consider training new token embeddings for the regression task? For example, instead of using digits 1-5, they could start from 5 untrained tokens. I would expect this to work worse, but I am curious about the results.

---

> ### Author Response · Authors · 2024-11-22
> **Response to Reviewer CzSf**
>
> > it would be worth slightly rewriting the paragraph after Lemma 1
>
> Thank you for the suggestion, we now incorporated it in the updated manuscript.
>
> > On the line 371, you are explaining that you are experimenting with a reduced training set for the STSB benchmark (using 1k examples instead of the whole training set). I understand that it is interesting to vary the amount of training data, but why are you only reducing the training set size on STSB and not on the other datasets?
>
> We selected STSB as one example regression dataset. To additionally analyze the impact of the data size, we add additional datasets: MovieLens and a synthetic dataset, thus expanding the range of dataset sizes (see Table 7 in the updated draft for the summary of all datasets and their corresponding sizes). Overall, we confirm RAFT improves over baselines across dataset sizes. Please see a common response to all reviewers “RAFT improvements over predictive head” for details.
>
> > Main weakness: While your paper studies regression tasks using autoregressive models, you observe on lines 465-467 that when training from scratch, the predictive head methods work best. Since encoder-only models such as BERT are widely used for tasks mentioned in the introduction (e.g. sentiment analysis, regression, ranking), it would be valuable to include a baseline of a large fine-tuned RoBERTa model on the same tasks that you present in the paper. Your findings would remain interesting and relevant in case the RoBERTa baseline was beating RAFT, but I think knowing how RAFT compares to fine-tuning RoBERTa is very relevant to practitioners. Therefore, I would suggest adding the following experiments (with the same hyperparameter tuning as RAFT):
> *Prediction head regression over a mean-pooling of RoBERTa representation (over sequence length)
> *Prediction head regression over the output for the CLS token of RoBERTa model
> *Both variants with either frozen RoBERTa weights, as well as unfrozen weights
>
> Please see our answer to this question in the common response to all reviewers: “Additional baselines”.
>
>
> > There could be more practical details on the fine-tuning process. Are you updating all the parameters of the model or did you freeze some of them (eg. update only the last linear layer)? Which optimizer are you using? If you are using Adam, what values of the betas, weight decay, and epsilon are you using? What batch size are you using? How long does one training run takes on average? How many machines did you need? The answers to these questions are important to help other researchers reproduce your results.
>
> We added a clarification regarding the settings in Appendix C. We update all parameters during the fine-tuning. We summarize specific settings below:
> * DROPOUT_RATE=0.1.
> * BATCH_SIZE=16.
> * We train for 200k steps and select the best step using the held out validation set (see Table 7 for details on the train/test/validation splits).
> * We use a constant learning rate schedule. We select the learning rate value over the validation set from the set of values: 1e-4, 1e-5, 1e-6.
> * We use the Adafactor optimizer to save memory during the fine-tuning (we find Adam to not perform better). The parameters for Adafactor are: epsilon1 = 1e-30, epsilon2 = 1e-3, decay_rate =0.8.
> * Each run takes approximately several hours on a TPU chip. We note that RAFT brings no additional overhead over standard fine-tuning, as it doesn’t introduce any new parameters, and it doesn’t slow down the training.
>
>
> > I don't understand why the authors are comparing MALI (regression on continuous values) against the Minimum Bayes Risk (MBR) prediction literature, since it optimizes non-regression metrics on the discrete grid (section 3.4). (...) It would be helpful to either strengthen the connection between MBR and MALI/RAFT, or consider moving this section to the appendix if it's not central to the paper's main argument. If you choose to keep it in the main text, consider explaining more clearly why this comparison is important for understanding RAFT's contributions.
>
> Our aim with this comparison was to additionally strengthen the motivation for the RAFT formulation by contrasting it against an MBR-style objective. We agree that the comparison between RAFT and MBR is non-central to the paper, and following the suggestion from the reviewer, we now delegated this discussion to the Appendix D in the updated manuscript.
>
> > Similar to the experiment presented in Table 11, did the authors consider training new token embeddings for the regression task? For example, instead of using digits 1-5, they could start from 5 untrained tokens. I would expect this to work worse, but I am curious about the results.
>
> Thank you for this suggestion. We include this ablation in Table 19 in the updated draft. We confirm the reviewer's intuition that it underperforms compared to when using digits 1-5.

---

> > ### Comment · Reviewer_CzSf · 2024-11-24
> >
> > I appreciate the authors efforts to address my questions. The presentation has now improved and makes the advantages of RAFT clearer. After considering the responses and other reviews, I have decided to increase my score.

---

### Official Review · Reviewer_Tf6L · 2024-11-01

**Soundness:** 3
**Presentation:** 4
**Contribution:** 3
**Rating:** 8
**Confidence:** 3

**Summary:**

This paper provides a comparison among several methods for autoregressive regression with LLMs, where the goal is to predict numerical value given the text input (e.g., text similarity). The authors propose a new fine-tuning approach, namely RAFT, which computes squared loss directly on the grid of possible outcomes without explicit sampling from the model. Empirical results show the effectiveness of the proposed approach across various datasets.

**Strengths:**

- Paper (in general) is well-written and easy to follow (even though there are still small things that require further clarification)
- Simple and effective idea
- Shown results indicate improvements across different tasks
- Additional analysis on several aspects (grid, pre-training, masking)

**Weaknesses:**

The predictive head approach closely follows the RAFT; however, there is no extended discussion of what sets RAFT apart.

Note that the below weaknesses are mostly minor.

The authors highlight the issues of the decoder-based LLMs. However, there is no baseline comparison (i.e., it is not clear what the potential upper bound is with current technologies). In the literature, performance for datasets like SSTB is often reported as Pierson/Spearman correlation
- Since some things are still unclear to me in the experiment setup (please see question), I am hesitant to assign a score

**Questions:**

- In Table 3, you report wireless predictive head results for the 2B model as 0.51, while in Table 4, the best result is 0.48 (special logits token). --
- In Table 3, the predictive head closely follows the results of RAFT. Could you please elaborate more on that? (more than described in 429-430)
- While comparing the effect of the grid, why have you excluded the STSB benchmark? It's a most interesting case since the grid 1-5 is a subset of the output space. I would be curious to see results and see if adding 0 makes any difference, for example

---

> ### Author Response · Authors · 2024-11-22
> **Response to Reviewer Tf6L**
>
> > The predictive head approach closely follows the RAFT; however, there is no extended discussion of what sets RAFT apart. (e.g. Table 3 and  429-430).
>
> Please see a common response to all reviewers “RAFT improvements over predictive head”.
>
> > The authors highlight the issues of the decoder-based LLMs. However, there is no baseline comparison (i.e., it is not clear what the potential upper bound is with current technologies). In the literature, performance for datasets like SSTB is often reported as Pierson/Spearman correlation
>
> We answer this question in the joint response to all reviewers: “Additional baselines”.
>
> > In Table 3, you report wireless predictive head results for the 2B model as 0.51, while in Table 4, the best result is 0.48 (special logits token). –
>
> We would like to clarify that 0.48 in Table 4 corresponds to the “+ 2-layer MLP” variant. The predictive head variant instead scores 0.51 in both Table 4 and Table 3.
>
> > While comparing the effect of the grid, why have you excluded the STSB benchmark? It's a most interesting case since the grid 1-5 is a subset of the output space. I would be curious to see results and see if adding 0 makes any difference, for example
>
> Per reviewer request, we include additional ablations of the grid for STSB in Table 19. For example, we find that only using token '5' in RAFT performs as well as using tokens '1'-'5'.
>
> We would like to clarify that adding 0 doesn’t make a difference to the loss, because in the RAFT rule, we multiply the probability by the digit, and so p(0|x) would get multiplied by 0.
>
> > Since some things are still unclear to me in the experiment setup (please see question), I am hesitant to assign a score
>
> Please do not hesitate to let us know if you have further questions we could clarify.

---

> > ### Comment · Reviewer_Tf6L · 2024-11-26
> > **Increase score to 8**
> >
> > Thank you for providing requested baselines and comparisons. I appreciate the work done by the authors. After careful consideration and reading other reviews/responses, I decided to increase my score.

---

### Official Review · Reviewer_7cP3 · 2024-11-03

**Soundness:** 3
**Presentation:** 4
**Contribution:** 3
**Rating:** 8
**Confidence:** 4

**Summary:**

The paper addresses regression tasks for language models, specifically using decoder transformer models to predict a numerical value $y$ based on an input sequence $x$. In this setup, two standard methods for regression are examined: (i) quantizing $y$ into discrete bins, where values are mapped to tokens (e.g., sentiment values represented by tokens 1, 2, 3, 4, 5) or (ii) training a multilayer perceptron (MLP) head on top of a decoder language model with an $L_2$ loss.

Each of these approaches has known limitations. Quantization lacks sensitivity to numerical proximity, treating all incorrect tokens as equally wrong (e.g., predicting 4 is treated the same as predicting 5, regardless of proximity to a true value of 3). Meanwhile, an MLP head with $L_2$ loss does not fully leverage the language model’s pretrained token prediction capabilities. To address these issues, the paper studies the MALI estimator, which uses a grid of tokens and calculates $\sum_{y \in \mathbf{y}_{\text{grid}}} p(\text{str}(y) | x ) . y $

Although MALI typically serves as a post-training estimator, the authors explore its application in fine-tuning language models for regression tasks, providing extensive comparisons and ablation studies to validate its performance against existing methods.

**Strengths:**

The paper is well-organized, with a clearly defined objective and comprehensive experimental comparisons, making it accessible and straightforward to follow. I enjoyed reading the paper.

**Weaknesses:**

1. **Performance Drop on Alternative Grids (Lines 902–903 and L531)**: You note a performance drop when using a grid of letters a–e, yet Table 11 shows similar MSE to the numerical grid. Could you clarify this inconsistency?

2. **Lemma 1 (Line 187)**: Based on the appendix, does the norm in Lemma 1 refer to an $L_1$ norm rather than $L_2$? The crafted example in the appendix (where a probability mass of $\epsilon / 2$ is positioned at 0 and $y_\text{max}$ implies that the $L_1$ loss should satisfy $E_x [ \| P(. | x)  - p(.|x)  \| ] <= \epsilon $ (the norm being L1), given that line 747 defines $ | \mathbb{P}(. | x) - p(.|x) |_{1} < \epsilon $.

3. **Token Granularity and Grid Complexity**:
   In the RAFT experiments, was the grid composed solely of single tokens? If so, does this also apply to the months-named grid?
   If the pretraining hypothesis holds that a simple grid (like 1–5) leverages pretrained information effectively, would a more complex grid (e.g., values such as 0.97, 2.3, 3.001, etc.) yield similar performance? I think this would be a very important question that can substantially strengthen the paper.
4. **Dataset Examples**: Including a table of dataset examples and inputs/outputs of the RAFT model would further help with clarity. It would help in verifying dataset characteristics too.

**Questions:**

1. **MSE Variations on Different Grids**: The MSE increase when using a months-named grid (versus alphabetic grids) is curious. Are the months tokenized as single or multiple tokens? Could this affect pretraining-based semantics, or could it be due to the number of tokens in use? Understanding why alphabet tokens retain performance while month names don’t could provide insight.
2. **Post-Training Distribution of the MALI Predictor**: After RAFT training, what does the MALI predictor’s distribution look like over grid elements? Since Lemma 2 emphasizes the grid’s max and min values, it would be informative to know how the grid’s distribution evolves post-training, including entropy relative to a uniform distribution over tokens.
3. **Baseline Comparisons**: While the paper compares MALI-based methods with other similar approaches, are these baselines considered state-of-the-art for your chosen tasks like sentiment analysis or the review estimations? Could simpler models, such as random forests, achieve lower MSE on these tasks? In other words, the experiments provide a good comparison among the methods based on finetuning a pre-trained transformer, but I wanted to clarify if these MSEs are actually state of the art compared to other methods too.

---

> ### Author Response · Authors · 2024-11-22
> **Response to Reviewer 7cP3**
>
> We thank the reviewer for the positive review and thoughtful comments.
>
> > “grid of letters a–e, yet Table 11 shows similar MSE to the numerical grid” and “Lemma 1 (Line 187)”
>
> Thank you for catching both misspellings. We now corrected the text to reflect the Table. We also fixed the norm in the Lemma statement to refer to the l1 norm.
>
> > Dataset Examples: Including a table of dataset examples and inputs/outputs of the RAFT model would further help with clarity. It would help in verifying dataset characteristics too.
>
> We report additional details on prompts and target ranges in Table 6.
>
> > Post-Training Distribution of the MALI Predictor: After RAFT training, what does the MALI predictor’s distribution look like over grid elements? Since Lemma 2 emphasizes the grid’s max and min values, it would be informative to know how the grid’s distribution evolves post-training, including entropy relative to a uniform distribution over tokens.
>
> This is a great suggestion. To investigate this, we inspect the distribution of softmax probabilities over the grid of tokens from RAFT for the Gemma-2 2B model on STSB over test examples.
>
> We find that, while the error decreases with training, the entropy increases after RAFT training, as we show in Figure 2 in the updated draft. This corresponds to the model on average spreading the probabilities over tokens more than prior to fine-tuning, as shown for specific examples in Figure 3 in the updated draft. We posit this to be beneficial, and indeed, RAFT fine-tuning does not restrict uncertainty of the model, contrary to the MBR fine-tuning (compare Lemma 3 and Lemma 5).
>
>
> > Baseline Comparisons: While the paper compares MALI-based methods with other similar approaches, are these baselines considered state-of-the-art for your chosen tasks like sentiment analysis or the review estimations? Could simpler models, such as random forests, achieve lower MSE on these tasks? In other words, the experiments provide a good comparison among the methods based on finetuning a pre-trained transformer, but I wanted to clarify if these MSEs are actually state of the art compared to other methods too.
>
> We answer this question and provide more baselines in the joint response to all reviewers: “Additional baselines”.
>
>
> > Token Granularity and Grid Complexity: In the RAFT experiments, was the grid composed solely of single tokens? If so, does this also apply to the months-named grid? If the pretraining hypothesis holds that a simple grid (like 1–5) leverages pretrained information effectively, would a more complex grid (e.g., values such as 0.97, 2.3, 3.001, etc.) yield similar performance? I think this would be a very important question that can substantially strengthen the paper.
>
> > MSE Variations on Different Grids: The MSE increase when using a months-named grid (versus alphabetic grids) is curious. Are the months tokenized as single or multiple tokens? Could this affect pretraining-based semantics, or could it be due to the number of tokens in use? Understanding why alphabet tokens retain performance while month names don’t could provide insight.
>
> Indeed, all grids we analyzed (including the months) in our work are single tokens in the vocabulary.
>
> We do not have a clear answer as to why months perform worse than character tokens. Our main hypothesis is that the pretraining alignment has the largest impact, which can explain why digits perform the best overall, but does not explain why months perform the worst (the length of the targets does not explain this observation).
>
> We agree more complex grid values are a great direction for exploration to understand the pretraining hypothesis better. We would like to clarify that more complex grid values that the reviewer gave as examples (0.97, 2.3, 3.001, etc.) are not in the pre-training vocabulary. Evaluating such more complex grids is certainly an interesting suggestion we will prioritize in explorations. However, it requires engineering effort to implement properly; e.g. when running multiple autoregressive steps at training, we need to carefully reuse key-value caches to avoid repeated passes through all transformer layers when computing activations for multi-token grid elements. A naive implementation significantly slows down the training.

---

### Author Response · Authors · 2024-11-22
**Summary of the changes in the updated draft**

A detailed list of changes can be found below. For your convenience, we've highlighted these changes within the updated draft.
* New section 3.2 on the limitations of standard fine-tuning and standard decoding, including new Lemma 1.
* New discussion that at step zero RAFT corresponds to MALI, and thus provides a good start for fine-tuning (see section 4.2).
* New dataset details (Table 6 and 7), including the target range in Table 6.
* New results showing that RAFT converges faster than predictive head (Figure 1).
* Training details in Appendix C.
* Delegated the discussion of MBR to Appendix D in the updated manuscript (following the suggestion from reviewr CzSf).
* New results on synthetic data on the role of pre-training (section 5.3 “Role of pre-training”, Appendix E.4, Table 13).
* New results on a natural language regression dataset: MovieLens-1M (Appendix E.5, Table 14).
* Additional baselines from prior works and with predictive head over RoBERTa (Appendix E.6, Table 15).
* Additional analyses of RAFT: different choices for loss, normalization, token initialization (Appendix E.7, Tables 16-20).
* Analyses of the distribution over tokens from the learnt MALI predictor (Appendix E.8, Figure 2, Figure 3).
* Clarifications and improvements to text as suggested by reviewers.

---

### Author Response · Authors · 2024-11-22
**Common response to all reviewers**

Two common questions from the reviewers were:
* (i) whether additional baselines could be added (7cP3, Tf6L, CzSf),
* (ii) why predictive head closely follows the results of RAFT, and how these two models differ (7cP3, Tf6L, AkBL)

We answer them in turn below.

> (i) Additional baselines (Reviewer 7cP3, Tf6L, CzSf)

**We report results with additional baselines in Table 15 in Appendix E.6, and find RAFT continues to perform the best.**

In detail, we provide additional baselines with prediction head over RoBERTa representation for the input sequence. We include:
* mean-pooling and CLS token variants, with both frozen and unfrozen RoBERTa weights in the fine-tuning
* the SMART method (Jiang et al., 2019) from previous work.

Moreover, as noted by reviewer Tf6L, previous works report Pearson and Spearman on STSB, e.g. (Jiang et al., 2019). We thus report the following metrics: Pearson, Spearman and RMSE (where available).

> (ii) RAFT improvements over predictive head (Reviewer Tf6L, CzSf, AkBL)

**We provide additional arguments as to the value of RAFT over the predictive head baseline**.

**Empirical arguments**. To further address the question about the relative performance of RAFT and predictive head, we ran additional experiments on two datasets:
* the real-world MovieLens-1M movie recommendation dataset (**Table 14 in Appendix E.5** in the updated draft)
* a synthetic regression dataset from Vacareanu et al. (2024a), referred to as “the Original #1 dataset” by the authors (**Table 13 in Appendix E.4** in the updated draft)

In both cases, we again find RAFT to improve over predictive head and autoregressive methods. In the synthetic dataset, we corroborate our observation from the language regression task experiment that RAFT improves over the predictive head approach when initialized from a pre-trained checkpoint, and not when model weights are initialized randomly (see Table 4). This provides additional support for our hypothesis that the alignment of RAFT to the next-token prediction pre-training task is the underlying reason for its better performance over the predictive head.

Overall, we would like to note that RAFT does improve over the prediction head in the majority of cases (i.e., out of 10 settings in Table 3, RAFT improves over the predictive head on 8 cases, and matches it on 2 cases).

**Additional empirical arguments**. There are further practical benefits of RAFT compared to predictive head:
* at step zero it corresponds to MALI method (Lukasik et al., 2024), which provides a strong zero-shot performance,
* it empirically converges much faster (e.g., see Table 12 and Figure 1), likely due to providing good initialization for the fine-tuning (see for additional added discussion in section 4.2 in the updated draft),

Lastly, we would like to highlight the **conceptual benefit** of RAFT compared to predictive head: on a technical level, RAFT is aligned with the pre-training task of the next token prediction and thus utilizes a different predictor function (see discussion in Section 4.2).

---

### Meta-Review · Area_Chair_b7Li · 2024-12-19

**Metareview:**

The paper presents RAFT, a novel regression-aware fine-tuning approach for LLMs based on the Bayes-optimal decision rule.

*Strengths of the paper:*
- Well-written and easy to follow
- Introduces a novel and conceptually sound approach (RAFT) for regression with LLMs
- Provides thorough experimental comparisons across diverse datasets and models
- Includes additional analyses and ablations to better understand the method

*Weaknesses and missing aspects:*
- The improvement over the predictive head baseline is not always significant, and in some cases the predictive head performs better
- More discussion on when autoregressive approaches, predictive heads, or RAFT might be preferable could be useful
- Inclusion of additional baselines like fine-tuned RoBERTa models would help contextualize the results


Overall, the paper makes a solid contribution by introducing RAFT, a novel and well-motivated approach for regression with LLMs. The extensive experiments demonstrate the effectiveness of RAFT across multiple benchmarks. While there are some weaknesses and areas for improvement, the reviewers generally found the paper to be well-executed and worthy of acceptance, especially after the authors provided clarifications and additional analyses in response to the reviewer comments.

**Additional Comments On Reviewer Discussion:**

In response to the reviewer comments, the authors made several key changes and clarifications during the rebuttal period. They addressed misspellings, provided more dataset details and additional baselines, investigated the learned MALI predictor's output distribution, and gave a more detailed explanation of how RAFT improves over the predictive head approach. The authors also rewrote a paragraph for better clarity, moved a less central comparison to the appendix, and included an experiment using untrained tokens. Additionally, they clarified the broader relevance of the regression problem addressed in the paper and discussed the tradeoffs between autoregressive sampling, predictive heads, and the proposed RAFT method. These changes and discussions helped address the reviewers' concerns and strengthen the overall presentation and contribution of the paper.

---

### Decision · Program_Chairs · 2025-01-22

Accept (Spotlight)